# Spatio-temporal dynamics of speleothem growth and glaciation in the British Isles

Sina Panitz[1,2], Michael Rogerson[1], Jack Longman[1], Nick Scroxton[3], Tim J. Lawson[4], Tim C. Atkinson[5], Vasile Ersek[1], James Baldini[6], Lisa Baldini[2], Stuart Umbo[1], Mahjoor A. Lone[1], Gideon M. Henderson[7], Sebastian F.M. Breitenbach[1]

[1]Department of Geography and Environmental Sciences, Northumbria University, Newcastle, NE1 8ST, UK
[2]School of Health & Life Sciences, Teesside University, Middlesbrough, TS1 3BX, UK
[3]Irish Climate Analysis and Research Units, Department of Geography, Maynooth University, Ireland
[4]School of Geosciences, University of Aberdeen, AB24 3UE, UK
[5]Department of Earth Sciences, University College London, WC1E 6BS, UK
[6]Department of Earth Sciences, Durham University, Durham, DH1 3LE, UK
[7]Department of Earth Sciences, University of Oxford, OX1 3AN, UK

*Correspondence to*: Sina Panitz (sina.panitz@gmail.com)

**Abstract.** Reconstructing the spatio-temporal dynamics of glaciations and permafrost largely relies on surface deposits and is therefore a challenge for every glacial period older than the last due to erosion. Consequently, glaciations and permafrost remain poorly constrained worldwide before c. 30 ka. Since speleothems (carbonate cave deposits) form from drip water and generally indicate the absence of an ice sheet and permafrost, we evaluate how speleothem growth phases defined by U-series dates align with past glacial-interglacial cycles. Further, we make the first systematic comparison of the spatial distribution of speleothem dates with independent reconstructions of the history of the British-Irish Ice Sheet (BIIS) to test how well geomorphologic ice reconstructions are replicated in the cave record. The frequency distribution of 1,020 U-series dates based on three different dating methods between 300 and 5 ka shows statistically significant periods of speleothem growth during the last interglacial and several interstadials during the last glacial. A pronounced decline in speleothem growth coincides with the Last Glacial Maximum before broad reactivation during deglaciation and into the Holocene. Spatio-temporal patterns in speleothem growth between 31 and 15 ka agree well with the surface-deposit-based reconstruction of the last BIIS. In data-rich regions, such as northern England, ice dynamics are well-replicated in the cave record, which provide additional evidence about the spatio-temporal distribution of permafrost dynamics. Beyond the Last Glacial Maximum, the distribution of speleothem dates across the British Isles offers the opportunity to improve chronological constraints on past ice sheet variability, with evidence for a highly dynamic Scottish ice sheet during the last glacial. The results provide independent evidence of ice distribution complementary to studies of surface geomorphology and geology, and the potential to extend reconstructions into permafrost and earlier glacial cycles. Whilst undersampling is currently the main limitation for speleothem-based ice and permafrost reconstruction even in relatively well-sampled parts of the British Isles, we show that speleothem dates obtained using modern mass spectrometry techniques reveal a higher spatio-

temporal resolution of glacial-interglacial cycles and glacial extent than previously possible. Further study of leads and lags in speleothem growth compared to surface deposition may provide new insights into landscape-scale dynamics during ice sheet growth and retreat.

## 1 Introduction

The maximum extent and retreat dynamics of the British-Irish Ice Sheet (BIIS) during Marine Isotope Stage (MIS) 2 are well
constrained by the BRITICE-CHRONO reconstruction (Clark et al., 2022a). However, geomorphological and sedimentological evidence of the growth dynamics of this ice sheet is much sparser, as it has been lost to erosion (Gibbard and Clark 2011; Clark et al., 2022a). Ongoing erosion becomes even more problematic for attempts to establish the history of glaciations and permafrost distribution older than the last. A detailed understanding of spatio-temporal variability of British-Irish glaciation and permafrost is crucial for refining model reconstructions of glacial climate variability in maritime
NW Europe. This knowledge enhances our ability to better forecast changes in retreating ice sheets, such as on the Antarctic Peninsula (Hughes et al., 2014), making these reconstructions particularly invaluable for predicting future ice sheet behaviour. Improving the chronological framework of both the BIIS and permafrost dynamics is equally important for regional palaeoecology and archaeology focussed on potential human habitation of Britain and Ireland. Consequently, the reconstruction of this highly dynamic glacial system remains a priority for regional Quaternary research.


Speleothems (secondary cave carbonates) are high-fidelity palaeoenvironmental archives capable of providing long, unaltered, and precisely dateable records of environmental changes (Wong and Breecker, 2015; Baldini et al., 2021). The presence of speleothems is an unambiguous sign for the absence of an ice sheet and/or continuous permafrost, as speleothem formation requires the inflow of fluid water. Vadose speleothem growth, which occurs within air-filled caves, is driven by
the supply of liquid water, saturated with dissolved inorganic carbon that precipitates in the cave environment. Infiltrating water is enriched in $CO_2$ that originates from microbial activity in the soil and dissolved carbon from the soil and host rock (Fairchild and Baker, 2012). Therefore, the growth of vadose speleothems implies the presence of both liquid water and (at least rudimentary) soil, and the absence of continuous ice. Vadose speleothems cannot form beneath ice sheets if the glacier base is frozen to their beds, thereby limiting the availability of liquid water, or if meltwater has filled the underlying caves
with ice or water (Dreybrodt, 1982). Thus, in polar and alpine climates zones, growth hiatuses can be linked to (peri-)glacial conditions and the formation of continuous permafrost (Baker et al., 1993; Vaks et al., 2020; Biller-Celander et al., 2021). Failure of water infiltration into caves could also arise from drought but given the precipitation pattern in the mid-latitudes of the British Isles the complete lack of infiltration on multi-annual to decadal timescale due to drought is unlikely, whereas continuous permafrost and/or ice cover during glacial periods are known to have repeatedly occurred (Clark et al. 2022).
However, speleothems can form subglacially in air-filled ventilated caves beneath warm-based glaciers where liquid water is available and an alternative source of acidity, such as pyrite oxidation provides a means for carbonate dissolution (Atkinson

et al. 1983; Skiba et al., 2023; Spoetl et al., 2024). Equally, while vadose speleothems are unable to form in the presence of continuous permafrost above and/or around the cave (Atkinson et al., 1986; Vaks et al. 2020), growth is possible, albeit starkly limited by temperature and water supply in case of discontinuous permafrost (Biller-Celander et al., 2021). The confounding processes of alternative sources of acidity or discontinuous permafrost could potentially alter the spatio-temporal links between the presence of permafrost and/or ice sheets and regional speleothem hiatuses.

Previous research has shown that British Isles speleothem growth aligns with the expected temporal pattern for the last glacial cycle (Atkinson et al., 1978, 1986; Gordon et al., 1989; Kashiwaya et al., 1991; Baker et al., 1993), with more abundant growth during warmer phases (interglacials and interstadials). This implies that in the temperate climate of Britain and Ireland, the presence of dated speleothems reflects the presence of soil and vegetation above the cave and the absence of either an ice sheet or continuous permafrost at that time and place. Since the early compilations of speleothem-based U-series dates, a vast number of additional dates have become available that improve the spatial and temporal coverage across the British-Irish Isles (e.g., Lundberg and McFarlane, 2007; Fankhauser et al., 2016). Moreover, major advances in uranium-series (U-Th) dating techniques offer more precise age control (Wendt et al., 2021) and thus a higher temporal resolution of past speleothem growth phases. For example, new speleothem dates from Scotland have revealed previously unrecorded interstadials in the British Isles that correlate with Greenland interstadials 14 and 12 and suggest a highly dynamic glacial system (Lawson et al., 2023). Consequently, it is timely to re-evaluate previous compilations of U-series dates, which were obtained using now discontinued dating methods that were once considered state-of-the-art (Li et al., 1989). This re-evaluation should incorporate new data and uncertainties regarding the assumption that speleothem growth is incompatible with permanent ice at the surface. Equally significant, the recent BRITICE-CHRONO (Clark et al., 2022a) reconstruction provides a fully independent dataset against which to test the ability of speleothems to reconstruct the spatio-temporal pattern of ice presence and absence.

Here, we collate all available U-Th dates of British and Irish speleothems with the aim of (1) assessing the timing of speleothem growth phases and glacial-interglacial cycles based on the different dating methods, and (2) integrating the new synthesis with independent evidence for late Quaternary glaciations from the BRITICE-CHRONO project to determine whether the spatial distribution of speleothem growth can further constrain the extent of the BIIS and older glaciations. This is the first time the spatio-temporal distribution of speleothems is compared in detail with the growth and decay of a former ice sheet.

## 2 Methods

We compiled 1,237 speleothem U-Th dates with finite uncertainties from 108 sites across Britain and Ireland from the literature. The dates were obtained by three radiometric dating methods: 699 using alpha spectrometry uranium series (ASU)

dating, 246 using thermal-ionisation mass spectrometry (TIMS) and 290 using multi-collector inductively coupled plasma

mass spectrometry (MC-ICP-MS). Here we separately analyse the speleothem growth phases obtained using the older and largely discontinued ASU dating method from the newer, more precise TIMS and MC-ICP-MS methods (Wendt et al., 2021). Published U-Th ages are used instead of recalculating ages from their measured $^{230}$Th/$^{234}$U/$^{238}$U ratios (e.g., Gordon et al., 1989), because the isotopic ratios are not always published and age uncertainties approach a Gaussian distribution for younger (<300-350 ka) ages (Weij et al., 2020). We are also aware that the calculation and correction of ages has changed in

the past 30 years, but recalculation for many ages is not possible, and we prefer to maximise the data density for this work. When published, the age corrected for initial $^{230}$Th was used, and any samples without dates, with infinite dates or infinite uncertainties were removed from the analysis. For ASU and TIMS speleothem dates that were published without uncertainties, we estimate 2σ errors based on the average uncertainty of the dating methods using the remainder of our dataset up to 500 ka (average uncertainties: ASU: ±17.65 %; TIMS: ±4.26 %). The average 2σ age uncertainty for MC-ICP-

MS dates is ±5.75 %.

The number of available U-Th dates declines exponentially with age, whilst dating uncertainties increase. We limit our in-depth analysis to the last 300 ka as a consequence (see also Gordon et al., 1989; Scroxton et al., 2016). We also remove ages where the sum of the age and upper age uncertainty exceeds 300 ka. This removes 48 ASU, and 28 TIMS and 41 MC-ICP-

MS dates. In addition, 28 ASU, 51 TIMS and 19 MC-ICP-MS dates with ages <5 ka were excluded from further analysis to reduce sampling bias as historically, active speleothems were generally not collected for conservation reasons and sampling is often done with the intention of collecting older material, resulting in a reduced number of samples available towards present (Gordon et al., 1989). This leaves a total of 1,020 dates (623 ASU, and 167 TIMS and 230 MC-ICP-MS dates) from 99 sites that cover the period 300 to 5 ka across Britain and Ireland (Fig. 1). In the TIMS and MC-ICP-MS dataset, Crag

Cave, SW Ireland, is overrepresented with a total of 104 dates that represent 26% of all measurements in this dataset (McDermott et al., 1999; Fankhauser et al., 2016). To remove this oversampling effect, we adjusted the total of 84 dates from Fankhauser et al. (2016) to 37 by removing ages with uncertainties overlapping older ages to ensure that the oldest available speleothem age of any time period is retained. Both the dataset including all dates available from Crag Cave and the bias-adjusted dataset are considered in the analysis, so our findings can be considered with and without this step.

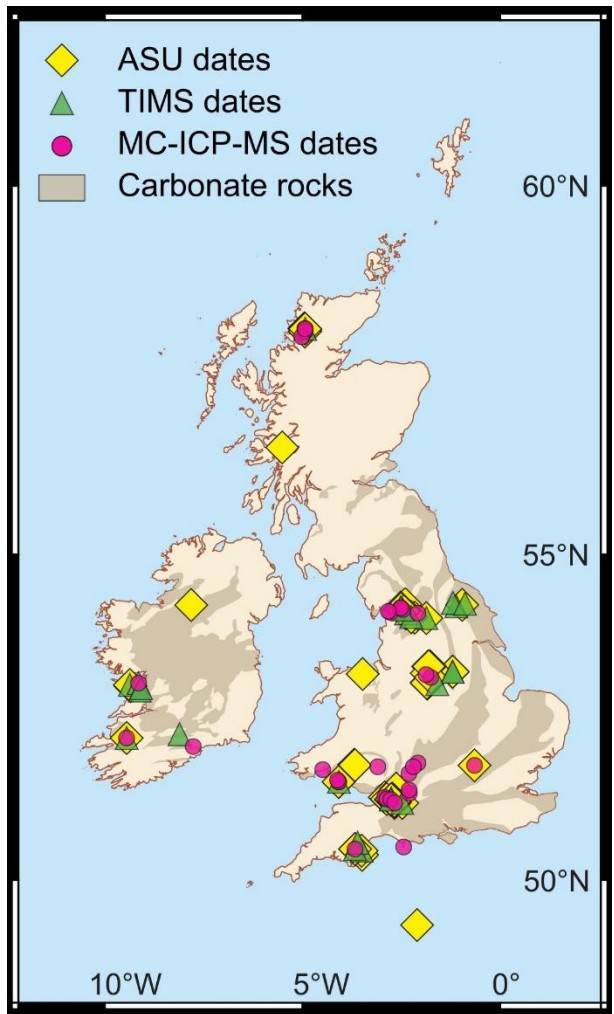

**Figure 1: Cave sites with dated speleothem samples in the British Isles dated using three different methods. Distribution of carbonate rocks from World Karst Aquifer Mapping project (Chen et al., 2017; note: this does not show Cambro-Ordovician dolostones or Jurassic carbonate outcrops in the Highlands of Scotland, and includes Cretaceous chalk).**

In addition to the visual inspection of the temporal distribution, speleothem growth phases were statistically analysed via construction of a probability density function (PDF). The PDF analysis was conducted on the full dataset, including the dates from all three methods, and separately on the ASU dates only (ASU dataset), and the combined TIMS & MC-ICP-MS dates only (MS dataset) (Fig. 2A-C). For the MS dataset, the PDF was calculated twice, first including all dates from Crag Cave and then with the reduced Crag Cave dataset (Fig. 2C). Relative age distributions were binned at 5-kyr resolution (Scroxton et al., 2016). The significance of the PDF peaks is dependent on increment size, with larger increments increasing the peak height, but reducing the certainty of the peak age, and smaller increments reducing the significance, but providing a more accurate age. As smaller increment bins (e.g., 0.5 ka) result in a very high inter-bin variability towards younger ages, they were not applied here.

To allow for better comparison of relative peak heights in the PDF, we removed the first-order exponential relationship seen in speleothem preservation and normalised the distribution using z-scores following Scroxton et al. (2016), but without the synthetic ages. For this, we fit an exponential function to the dataset of age versus frequency ($y = 13.388e^{-0.009x}$). Then, for each point in time we subtracted the expected value (i.e if the function fitted the data perfectly) from the observed value, thereby removing the underlying 'natural attrition' trend that reduces the height (depth)

of peaks (troughs) with time to allow for better comparison of relative peak heights. These values are then converted to standard scores (z-scores) to allow for the variability to be more easily visualised. This approach is based on the assumption that 1) the rate of decay of speleothems remains relatively constant over time periods longer than 105 years and 2) random sampling will over-represent the youngest (and uppermost) periods of growth. In the boreal to temperate climate of the British Isles, caves have been subject to periods of enhanced erosion, particularly during deglaciations, such that the 'constant' decay observed by Scroxton et al. (2016) in the more temperate to tropical zones may not apply. However, when

155

observed over numerous deglaciations, even this variability will not remove the underlying exponential 'attrition rate' (Fig. A1).

160

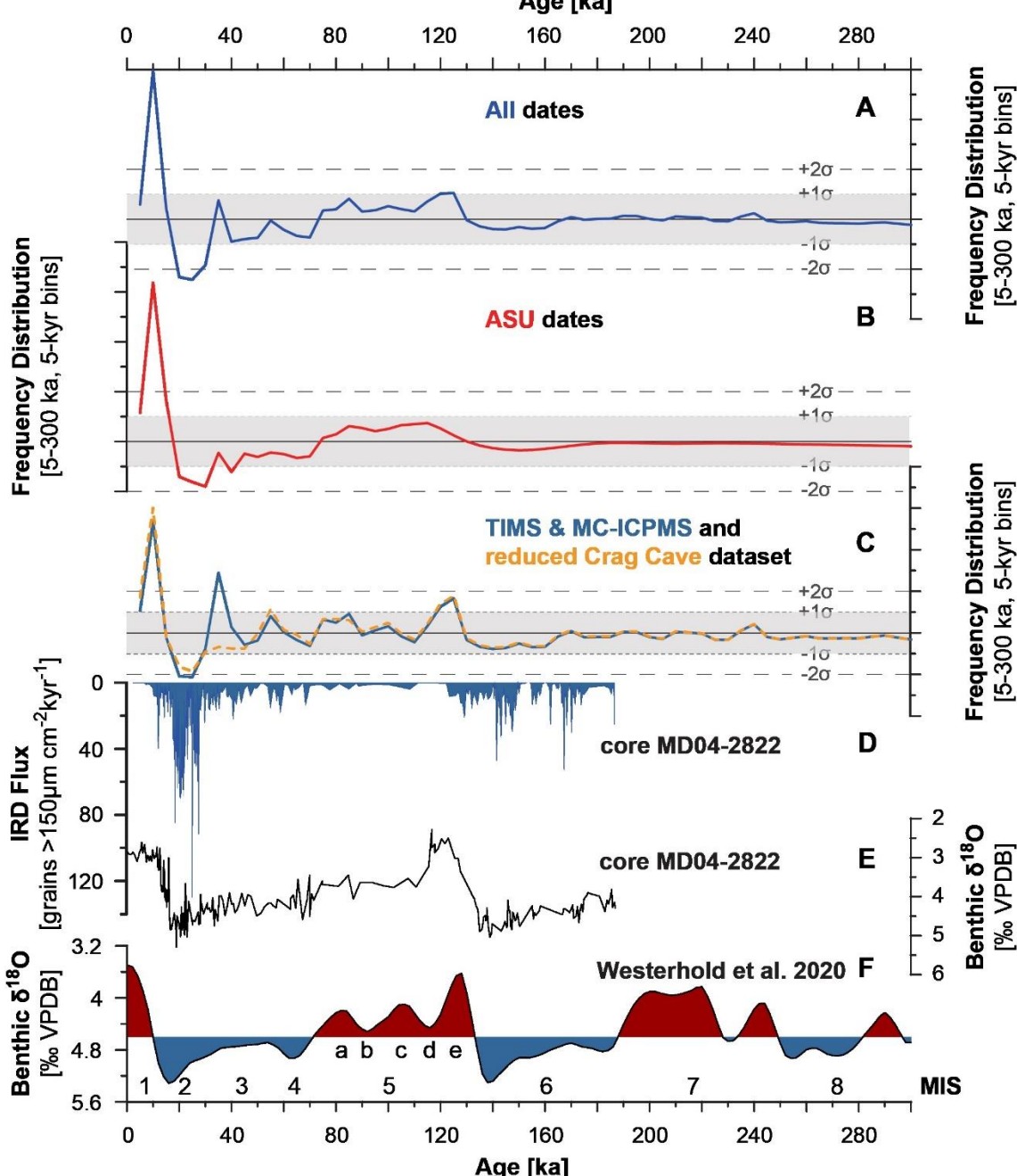

**Figure 2: (A-C)** Normalised frequency distributions of speleothem growth between 300 and 5 ka based on the different datasets in comparison to regional and global climate records: **(D)** ice-rafted debris (IRD) flux and **(E)** benthic $\delta^{18}O$ from site MD04-2822 in the NE Atlantic off the west coast of Scotland (Hibbert et al., 2010), and **(F)** benthic $\delta^{18}O$ from Westerhold et al. (2020) with numbers and letters denoting marine isotope stages and substages, respectively. **(A-C)** Grey horizontal lines mark zero (solid line), $1\sigma$ (dotted line) and $2\sigma$ (dashed line) standard deviations.

For comparison of age uncertainties between the two datasets (ASU and MS dates), dates were binned at 5-kyr resolution (i.e., the same as the PDF resolution) and the average uncertainty calculated in each bin (Fig. 3). Generally, age uncertainties increase with age for both methods, which is reflected in the estimates.

To prepare spatial time slices and for comparison to the empirical BIIS reconstruction at 1000-year intervals (Clark et al., 2022a), the speleothem dates were binned at 1-ka intervals, with 0.5 ka used as cut-off for bins. For example, speleothem dates between 14.50 to 15.49 were allocated to the 15 ka timeslice. The spatial distribution of pre-last glacial (>31 ka) growth periods is presented over longer time periods, grouping 1-ka time intervals based on notable changes in the distribution of speleothem growth across the British Isles.

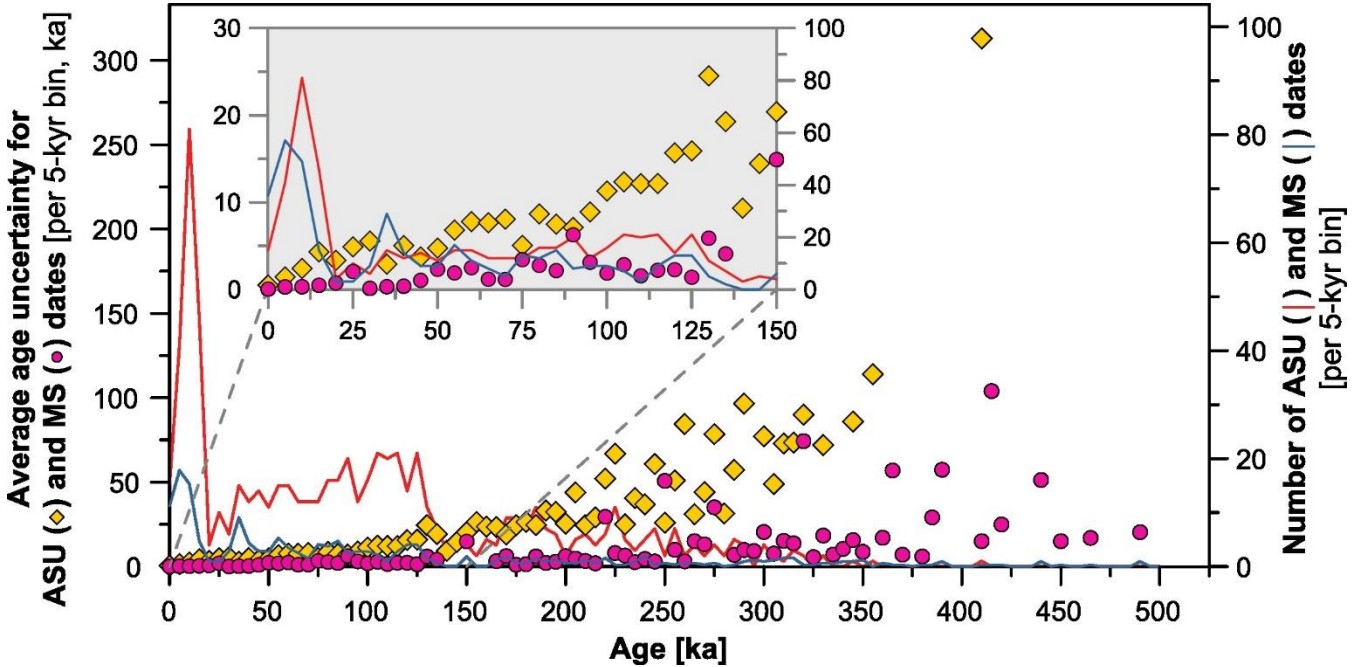

**Figure 3: Comparison of average uncertainties by method within 5-kyr bins and the number of dates for each method per 5-kyr bin. Mass spectrometry (MS) dates include TIMS and MC-ICP-MS) dates.**

## 3 Results

The PDF distributions of speleothem dates from the British Isles over the last 300 ka are shown in Fig. 2. The composite PDF speleothem growth curve based on all ASU, TIMS and MC-ICP-MS dates shows no discernable, statistically significant structure prior to 125 ka (Fig. 2A). A peak at 125–120 ka exceeds one standard deviation above the exponential baseline, a peak at 10 ka exceeds two standard deviations above the baseline, and one period at 25–20 ka exceeds two standard

deviations below the baseline. To the first order, peaks are indicative of enhanced speleothem growth in caves in the British Isles under interglacial conditions, and the troughs of reduced speleothem growth during glacial conditions.

    In the PDF of the separate ASU dataset, only the peak at 10 ka surpasses two standard deviations above the baseline, and a longer trough is found between 30 and 20 ka, with another minimum at 40 ka exceeding one standard deviation below

baseline. A prolonged peak from 120 to 80 ka remains below one standard deviation as does a trough between 70 and 45 ka. Whilst these maxima and minima suggest enhanced and reduced speleothem growth, respectively, they are not statistically significant (Fig. 2B).

    We find a higher speleothem growth variability in the MS dataset (Fig. 2B,C). Minor and statistically insignificant peaks are

recorded prior to 125 ka, with one notable peak at c. 240 ka. A clear and significant ($>1\sigma$) peak at 125 ka in the MS dataset shows that the ASU dates lack precision to resolve variability, with the average $2\sigma$ uncertainty being an order of magnitude higher in the ASU than the MS dataset in this time period (Fig. 3). Two other peaks at 85 and 55 ka almost reach one standard deviation above the exponential baseline. with two peaks at 35 and 10 ka exceeding two standard deviations. However, the peak at 35 ka can be ascribed to an overrepresentation of dates from Crag Cave. In the adjusted dataset (i.e.,

excluding selected Crag Cave dates) the peaks at 125 and 55 ka, and between 25-20 ka exceed one standard deviation above the baseline, and the peak at 10 ka surpasses two standard deviations above it (Fig. 2C).

## 4 Discussion

    The presence of a single accurate and reliable speleothem date at any given time and place is indicative of the absence of an ice sheet (unless subglacial growth can be ascertained) or continuous permafrost above the cave. Individual dates allow a

much higher degree of detail when estimating local to regional (peri)glacial conditions while a frequency curve (PDF record) only indicates the relative probability that a given date falls within an increment. Thus, PDF reconstructions are useful tools to highlight regional glaciation dynamics that can be compared to hemispheric or global climatic changes, or global climate model results, whereas individual dates give insights into local conditions.

### 4.1 Temporal distribution of speleothem growth

**4.1.1 Long-term speleothem growth variability**

    As the number of available dates decreases with age, the relative magnitude of peaks and troughs in the PDF record also declines, thereby losing significance. In the case of the British Isles, speleothem data covers a relatively large geographical area, spanning eight degrees latitude, and thus spatial and temporal variations in speleothem growth phases. This means that the transitions between and the magnitudes of peaks and troughs are smoothed as compared to analysing speleothem age

distributions from a single cave system or a confined geographical area (e.g., Ayliffe et al., 1998; Jo, et al., 2014; Scroxton et

al., 2016). To mitigate this effect on the reconstructions, here, PDF peaks are considered significant at the ≥1σ level. Despite the applied correction for speleothem preservation (Scroxton et al., 2016), we retain a clearer signal in the younger section of the composite records.

The speleothem growth curves reconstructed using the two MS and ASU sub-sets exhibit a common structure. In all PDF records, speleothem growth phases reflect the last glacial-interglacial cycle (Fig. 2A-C), which supports the interpretation that the presence/absence of speleothem growth is related to large-scale Quaternary glacial cycles in Britain and Ireland. However, compared to the ASU dataset, the PDF based on the MS dataset exhibits a higher temporal resolution, arising from the higher precision of mass spectrometry dating techniques as a result of dramatic analytical improvements over the last few

decades (Wendt et al., 2021). Consequently, the MS dates are able to resolve a clearer structure of glacial-interglacial cycles further into the past than the ASU dates. In both cases, however, the magnitudes of peaks and troughs are much lower prior to the last interglacial (MIS 5e).

The lack of statistically significant peaks in our speleothem growth curves before 130 ka almost certainly reflects

undersampling as a result of the natural attrition of speleothems as well as the relatively large age uncertainties associated with the available dates (Fig. A2). The period is covered by 187 ASU dates, and 63 TIMS and 80 MC-ICP-MS dates (27% of each dataset). However, a minor peak at 240 ka in the MS dataset is likely associated with interglacial MIS 7.5 and a prolonged minimum between 160 and 130 ka can probably be linked to the latter half of glacial MIS 6 (Fig. 2C; Lisiecki and Raymo, 2005). These two examples show the potential of constraining the timing of older glacial-interglacial cycles on the

British Isles using speleothem growth phases once more data becomes available. As each dated speleothem indicates favourable growth conditions at the cave site, the absence of ice and continuous permafrost at a location can still be inferred from dates from an individual site even when regional reconstruction is restricted by small datasets. Therefore, speleothem ages provide the opportunity to expand the reconstruction of ice and permafrost in the future. For both PDF and individual age approaches, increasing the number of MS dates for the period 300-130 ka is a priority for future investigation of past

glacial/interglacial climate variability.

### 4.1.2 The last glacial-interglacial cycle

During the last interglacial (MIS 5e), peaks in both the ASU (115 ka) and MS (125 ka) datasets indicate enhanced speleothem growth and warm climatic conditions across the British Isles (Fig. 2B,C). This is in line palaeontological evidence of warmer-than-present climatic conditions during MIS 5e (Candy et al., 2016). In the combined and ASU datasets,

speleothem growth remains relatively high until c. 85 ka before declining into glacial MIS 4 (Fig. 2A,B). In the MS dataset, a higher climate variability is observed, with two troughs (at 110 and 90 ka) and peaks (at 100 and 85-75 ka) coinciding with stadials MIS 5d and MIS 5b, and interstadials MIS 5c and MIS 5a, respectively (Fig. 2C), reflecting the higher precision of the MS dates. The addition of further data will likely allow further structure to emerge.

At c. 70 ka, the ASU dataset indicates a shift from sustained speleothem growth to a period of reduced growth, culminating with a minimum during the Last Glacial Maximum (MIS 2; Fig. 2B). Growth phases indicated in the MS dataset show a similar structure, but with more pronounced peaks at 85 and 55 ka, which possibly reflect interstadials. With the inclusion of the large number of dates from Crag Cave, the low-growth period is further restricted to the Last Glacial Maximum, but this is unlikely to be representative of the whole region. The difference between the PDF curves with all MS dates and the

reduced Crag Cave dataset illustrates the impact the number of available dates has on this kind of analysis.

The interpretation that speleothem growth dynamics follow ice sheet variability is further supported by the main pattern in the Ice Rafted Debris (IRD) record from the Atlantic just west of the BIIS (Hibbert et al. 2010). The record shows minimal IRD supply at times of significant speleothem growth (MIS 5e and Holocene), whilst maximal IRD supply aligns with a

minimum in speleothem deposition during the Last Glacial Maximum (Fig. 2C,D).

### 4.2 Spatial distribution of speleothem growth

#### 4.2.1 Speleothem growth and ice sheet coverage

To test the hypothesis that the distribution of speleothem growth can be used to constrain the extent of past glaciations, we compare the spatial distribution of speleothem dates to the empirical BIIS reconstruction in 1-ka intervals from 31 to 15 ka

(Fig. 4-5; Clark et al., 2022a). The empirical BIIS reconstruction based on surface deposits is presented as three possible ice sheet extents: minimum, optimum and maximum. The ice sheet model was nudged to try to fit the optimum ice sheet extent, but failing that, constrained by the minimum and maximum limits (Clark et al., 2022a).

In the ASU dataset, ten speleothem dates are recorded at eight sites across the British Isles between 31-29 ka and 27-25 ka,

partly overlapping with different ice sheet limits as indicated by the BIIS reconstruction (Fig. 4). The dates marginally overlapping with the minimum and optimum ice sheet limit may reflect a higher degree of ice sheet variability, but age uncertainties exceed 1 ka and hamper interpretation (Fig. 3).

In the MS dataset, the only site for which speleothem dates are available during early MIS 2 (31-25 ka) is Crag Cave in SW

Ireland (Fig. 4A-D). The majority of the available dates from this site have uncertainties of ±0.3 ka (Frankhauser et al., 2016), and very little evidence for detrital thorium contamination, which allows for direct comparison with the 1-ka BIIS reconstruction (Clarke et al., 2022). A short break in deposition at the site between 27 and 24 ka is consistent with ice coverage during maximum BIIS extent (Fig 4E-H). However, the BIIS reconstruction also suggests the presence of an ice sheet in SW Ireland at 29-28 ka as indicated by the optimum ice sheet limit, overlapping with speleothem growth at Crag

Cave. The Crag Cave record is interpreted as showing growth phases during Greenland Interstadials and not Greenland Stadials (Fankhauser et al., 2016), therefore suggesting a

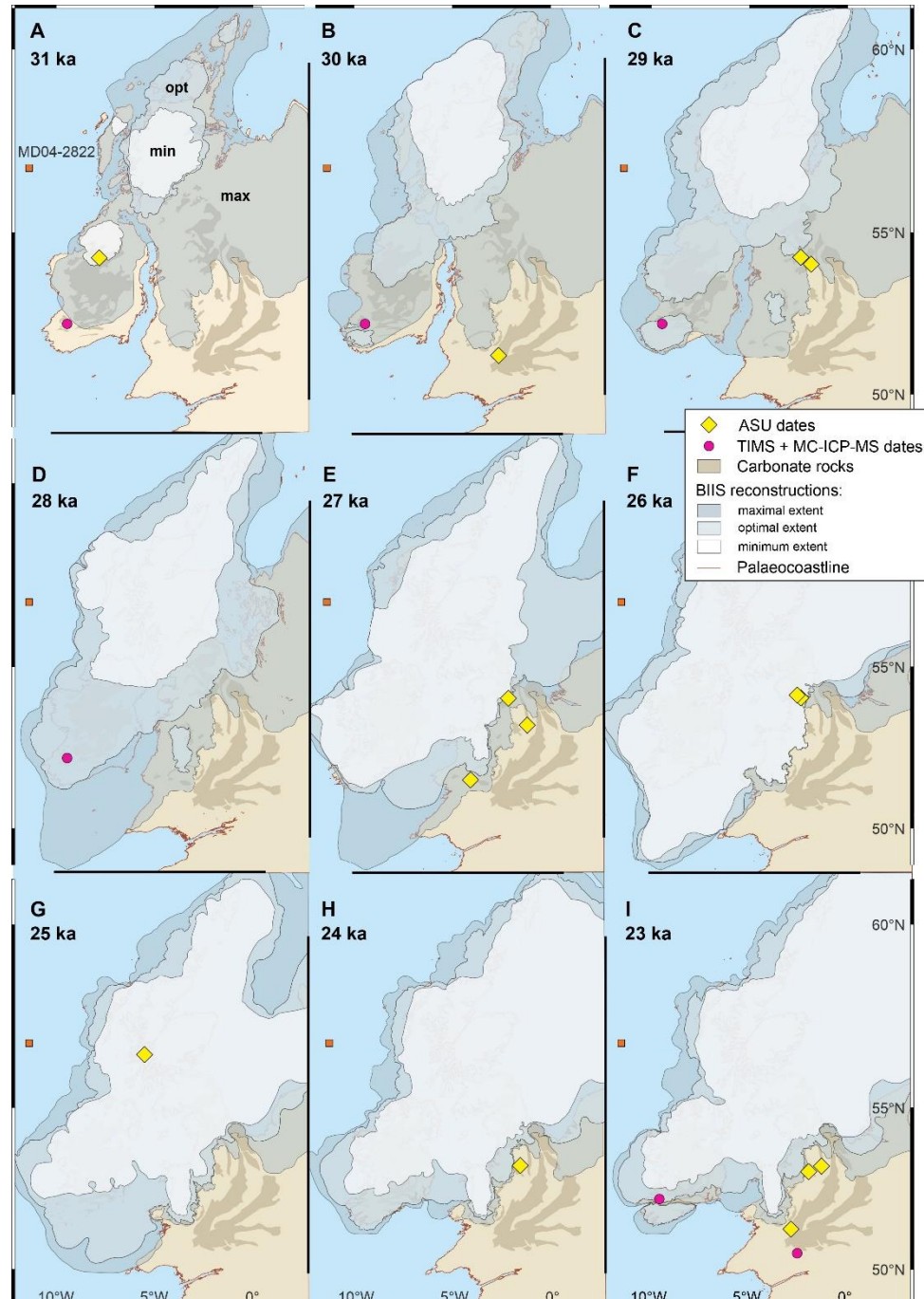

**Figure 4: Speleothem growth in the British Isles during the last glacial period and location of site MD04-2822 (Hibbert et al., 2010). Empirical British-Irish Ice Sheet reconstruction (BIIS) (minimal, optimal and maximum extent) and modelled position of palaeo-coastline from Clark et al. (2022b). Distribution of carbonate rocks from World Karst Aquifer Mapping project (Chen et al., 2017).**

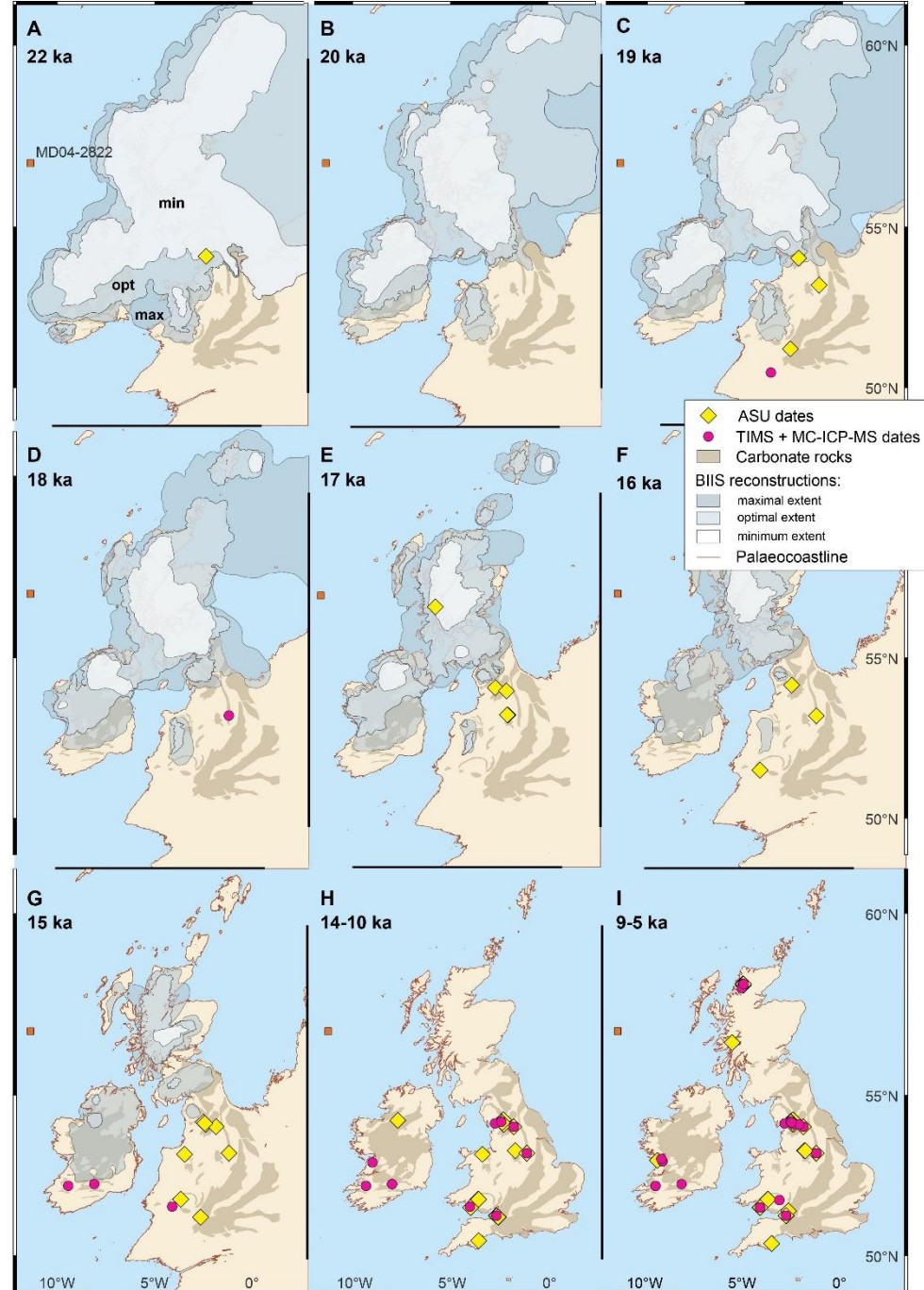

**Figure 5: Speleothem growth in the British Isles during the last glacial period and Holocene, and location of site MD04-2822 (Hibbert et al., 2010). (A-G) Empirical British-Irish Ice Sheet reconstruction (BIIS) (minimal, optimal and maximum extent) and modelled position of palaeo-coastline from Clark et al. (2022b). Distribution of carbonate rocks from World Karst Aquifer Mapping project (Chen et al., 2017).**

more dynamic ice-sheet margin during millennial-scale climate variability than can be captured in the 1-ka time steps of the largely luminescence-date constrained BIIS reconstruction (Fig. 4C,D; Clark et al., 2022a).


Following the maximum ice extent at 26 ka (Clark et al., 2022a), ice-marginal ASU-dates at 24-23 ka in northern England may be indicative of the absence of continuous permafrost in these periglacial environments (Fig 4H-I), but age uncertainties (2σ error = ±2-3 ka) must be reduced for a clearer interpretation. The first, more precise MS-dates are recorded at Crag Cave at 23 ka (2σ error = ±0.1-0.17 ka), indicating active infiltration at the site. Another MS date, but with high age uncertainties

(2σ error = ±5.9 ka), available from the Isle of Portland, southern England (Fig. 4I). The Crag Cave date overlaps with the proposed optimum extent of the BIIS, reflecting a highly variable ice margin or simply dating offsets. The onset of speleothem growth at Portland, which was not glaciated during the last glacial period (Clarke et al., 2022a), likely records the thawing of permafrost. An ASU date in northern England at 22 ka overlaps with the optimum BIIS reconstruction (Fig. 5A) but considering the relatively large age uncertainty (2σ error = ±2.4 ka), it could easily fall into a different time slice and

is not in conflict with the BIIS reconstruction. After 20 ka, most dates are located in BIIS reconstructed ice-free regions of the British Isles (Fig. 5). One apparent exception is an ASU date within the minimum ice sheet limits in western Scotland at 17 ka (2σ error = ±5.8 ka; Fig. 5E). A noticeable increase in the number of dates after 15 ka supports an ice-sheet confined to parts of Scotland and Northern Ireland as indicated by the BIIS reconstruction (Fig. 5).

In summary, the spatial comparison of speleothem growth and the BIIS reconstruction indicates that both the ASU and MS datasets are generally in strong agreement with the ice sheet reconstruction based on surface deposits (Clarke et al., 2022). The majority of ASU dates align with the more precisely resolved picture gleaned from the MS dataset. The few dates that conflict with the BIIS reconstruction are likely due to analytical uncertainties, and in the case of ASU dates, should be assessed by re-dating the samples using MC-ICP-MS dating techniques before concluding that the ice sheet reconstructions

contradict speleothem growth in these regions. In case of misalignments between the very thoroughly sampled Crag Cave site and the BIIS reconstruction, the apparent conflicts potentially reveal variability at the dynamic ice sheet margins not captured by surface geology as suggested by Fankhauser et al. (2016).

### 4.2.2 Lags in the onset of speleothem growth following ice sheet retreat

Whereas minimal conflicts between speleothem growth and the BIIS reconstruction exist, we do find a considerable lag

between reconstructed ice loss and speleothem growth initiation. The best characterised site, Crag Cave, suggests a lag of >0.11 ka (Fankhauser et al., 2016), but in central Ireland, northern England and Scotland, the lag between ice decay and speleothem growth initiation is generally 5-7 ka (Fig. 5). At Crag Cave, the rapid restart of speleothem growth during the Last Glacial Maximum may be explained by a fast re-establishment of vegetation or the storage of organic materials in frozen soils and possibly the epikarst zone of the cave (Fankhauser et al., 2016), in combination with basal ice melting at this

high-precipitation site (Alexander et al., 2011). Longer lags further to the north may be because of differences in the persistence of permafrost or in the initiation of soil formation after deglaciation, or simply the result of undersampling.

In NW Scotland, a pollen record with a basal age of c. 15 ka (2σ error = c. ±0.7 ka) is reported from an area exposed since 18 ka (Ranner et al., 2005; Clark at el., 2022a). This direct dating indicates that slow recovery of soil in regions experiencing
deeper glacial weathering than southwest Ireland is a reasonable explanation for the larger lag. However, the onset of pollen deposition within c. 3 ka after ice retreat is still inconsistent with the 5-7 ka lag observed in the speleothem dataset. Persistence of permafrost for a period after glacial withdrawal is a likely environmental cause for this additional delay, but some speleothems activated earlier after ice sheet retreat may not have been sampled. This is supported by new MS dates from NW Scotland, which record short-lived Greenland Interstadials 14 (c. 53 ka) and 12 (c. 46 ka) (Lawson et al., 2023),
suggesting that ice variability was very dynamic, with a relatively quick response in speleothem growth. This rapid response would be analogous to the way speleothem growth at Crag Cave behaved during the Last Glacial Maximum (Fankhauser et al., 2016). Consequently, we find it likely that some of the apparent lag arises from undersampling of regions outside of the exceptionally well investigated Crag Cave.

### 4.2.3 Regional case study: Northern England

A weakness of the regional PDF compilations is the eight degrees latitude difference between the northern and southern sites, and the consequent conflation of changes in the non-glaciated and highly glaciated parts of Britain and Ireland. Further insight can be gained by investigating one sub-region in detail for the Last Glacial Maximum and deglaciation. Northern England has a relatively high density of speleothem dates following ice sheet retreat and is thus suitable for deeper investigation at this stage.

Across this sub-region there is a prominent minimum of speleothem dates between c. 35 and 15 ka at 54°N and before c. 20 ka at 53°N. Dates within this minimum are from the ASU method, with MS dates absent between c. 36 and 14 ka at 54°N and c. 36 and 19 ka at 53°N (Fig. 6). The BIIS reconstruction suggests that ice was present from c. 28 to 19 ka in the northern part of the sub-region, but never reached the southern part except for the lower altitudes to the west between 27 and
22ka (Fig. 7; Clark et al., 2022a). This leaves five ASU dates in conflict with the ice sheet reconstruction, but a much wider gap in speleothem growth across the sub-region, which may be explained directly by the presence of permafrost around the ice sheet. Based on the available MS dates, there is a maximum lead of 7 ka between cessation of speleothem growth due to permafrost formation prior to ice sheet advance at 54°N, likely reflecting sampling bias (Fig. 6).

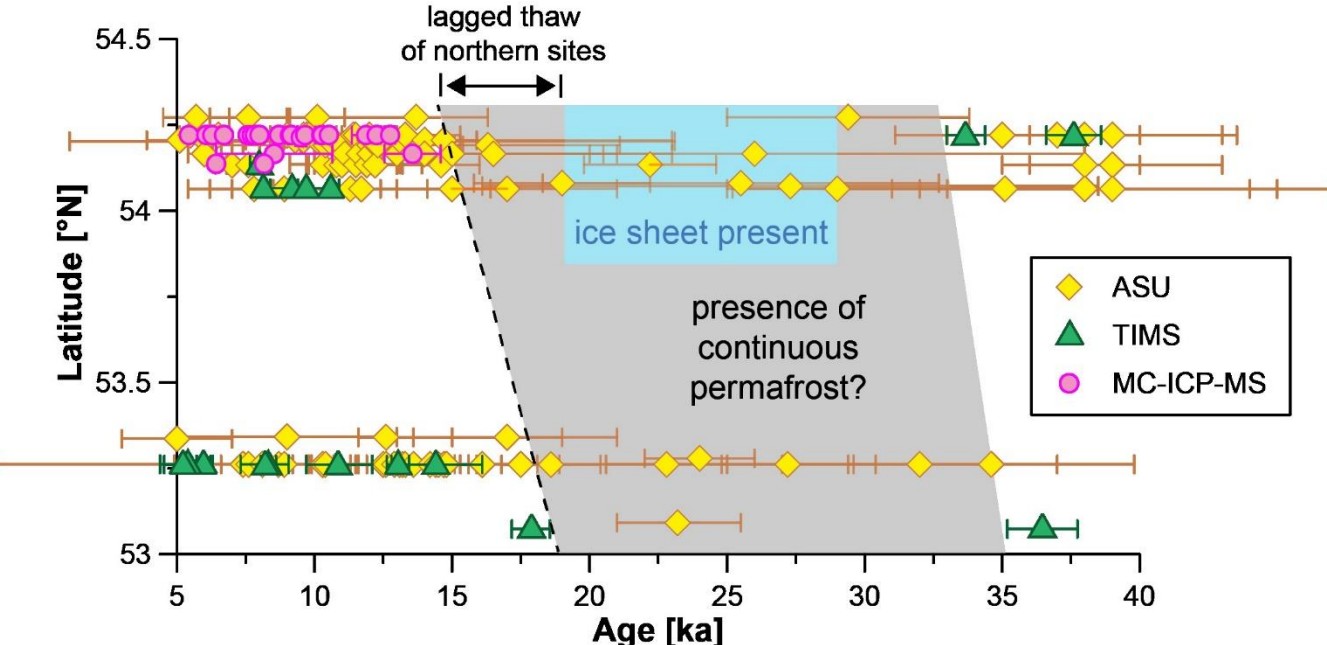

 **Figure 6: Distribution of speleothem ages in northern England between 40 and 5 ka based on the three different dating methods. Ice presence inferred from the empirical British-Irish Ice Sheet reconstruction minimal to optimal extent (Clark et al., 2022b). Inferred potential presence of continuous permafrost based on available TIMS and MC-ICP-MS dates.**

Following the retreat of the BIIS between 20 and 18 ka in northern England, the rapid return to speleothem growth at 53°N suggests continuous permafrost was lost in that region quickly. The first post-glacial speleothem MS date around 19 ka at 360 53°N suggests that growth may have recommenced in this region only centuries after ice retreat (Fig. 6). By 17-16 ka only isolated ice sheets are proposed to remain in northern England according to the BIIS reconstruction, which is in agreement with an increased number of speleothem dates recorded following deglaciation and the establishment of more favourable climatic conditions (Fig. 7). However, as with other glaciated regions across the Isles (discussed above), a considerable lag exists between ice retreat at 54°N and return to abundant speleothem growth (c. 3 ka for ASU dates and 5 ka for MC-ICP- 365 MS dates), possibly reflecting the presence of continuous permafrost following deglaciation (Fig. 6). A lag of max. 5 ka between permafrost thaw and the initiation of speleothem growth (Fig. 6). However, further dating of speleothems should refine the timing and nature of this permafrost-induced lead and lag between speleothem growth and ice advance and retreat, which is a component currently missing from the BIIS reconstruction.

### 4.2.4 Reconstructing glacial extent in periods older than the BRITICE-CHRONO reconstructions (130-31 ka)

The good agreement between the spatial distribution of speleothem dates and the empirical BIIS reconstruction for the period 31-15 ka suggests that the spatial pattern of speleothem growth may be used to reconstruct glacial dynamics beyond

the Last Glacial Maximum. Here, we use the speleothem compilation to constrain glacial variability across the British Isles between 130 and 31 ka.

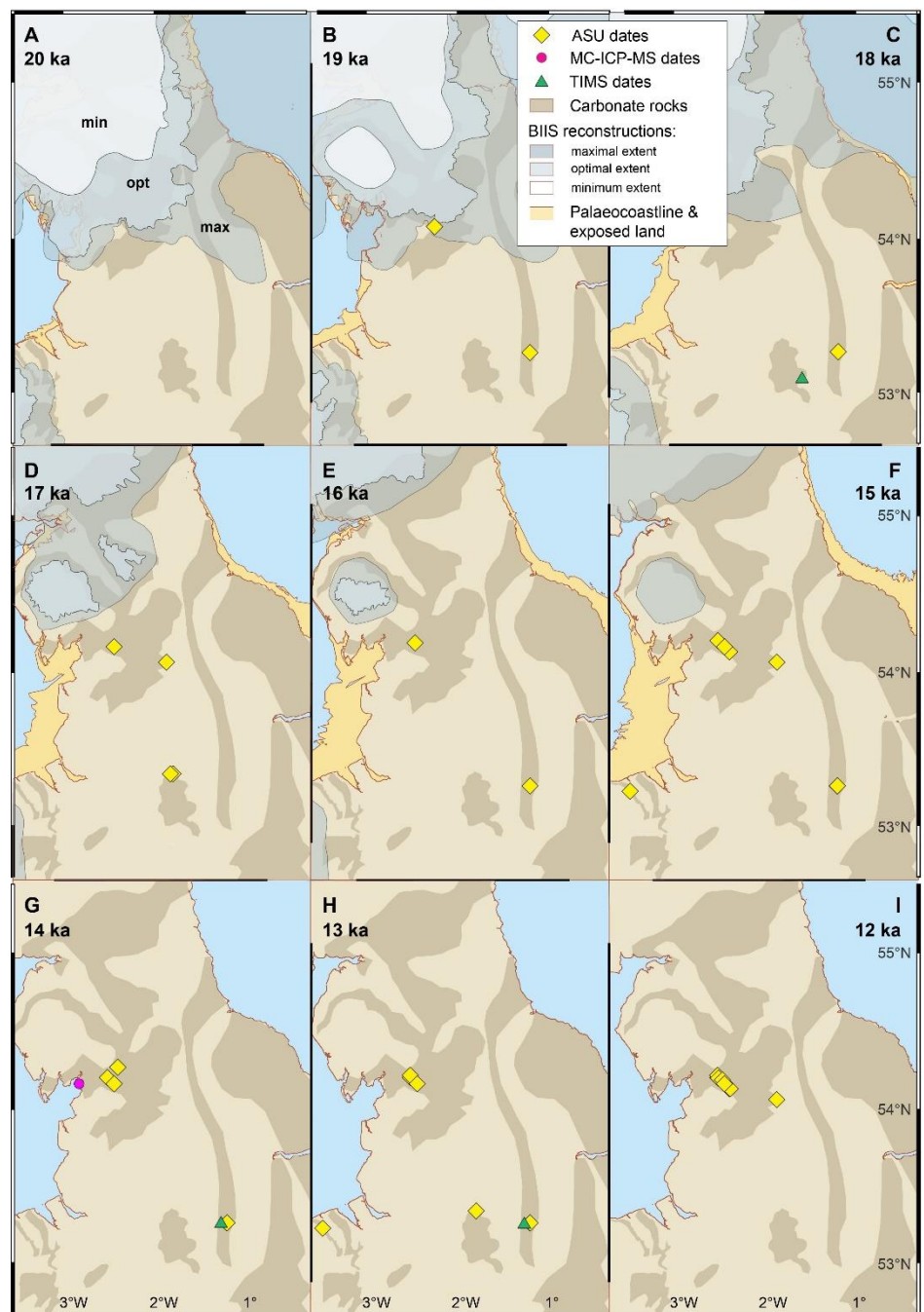

**Figure 7: Speleothem growth in northern England from 20 to 12 ka. (A-F) Empirical British-Irish Ice Sheet reconstruction (BIIS) (minimal, optimal and maximum extent) and modelled position of palaeo-coastline from Clark et al. (2022b). Distribution of carbonate rocks from World Karst Aquifer Mapping project (Chen et al., 2017).**

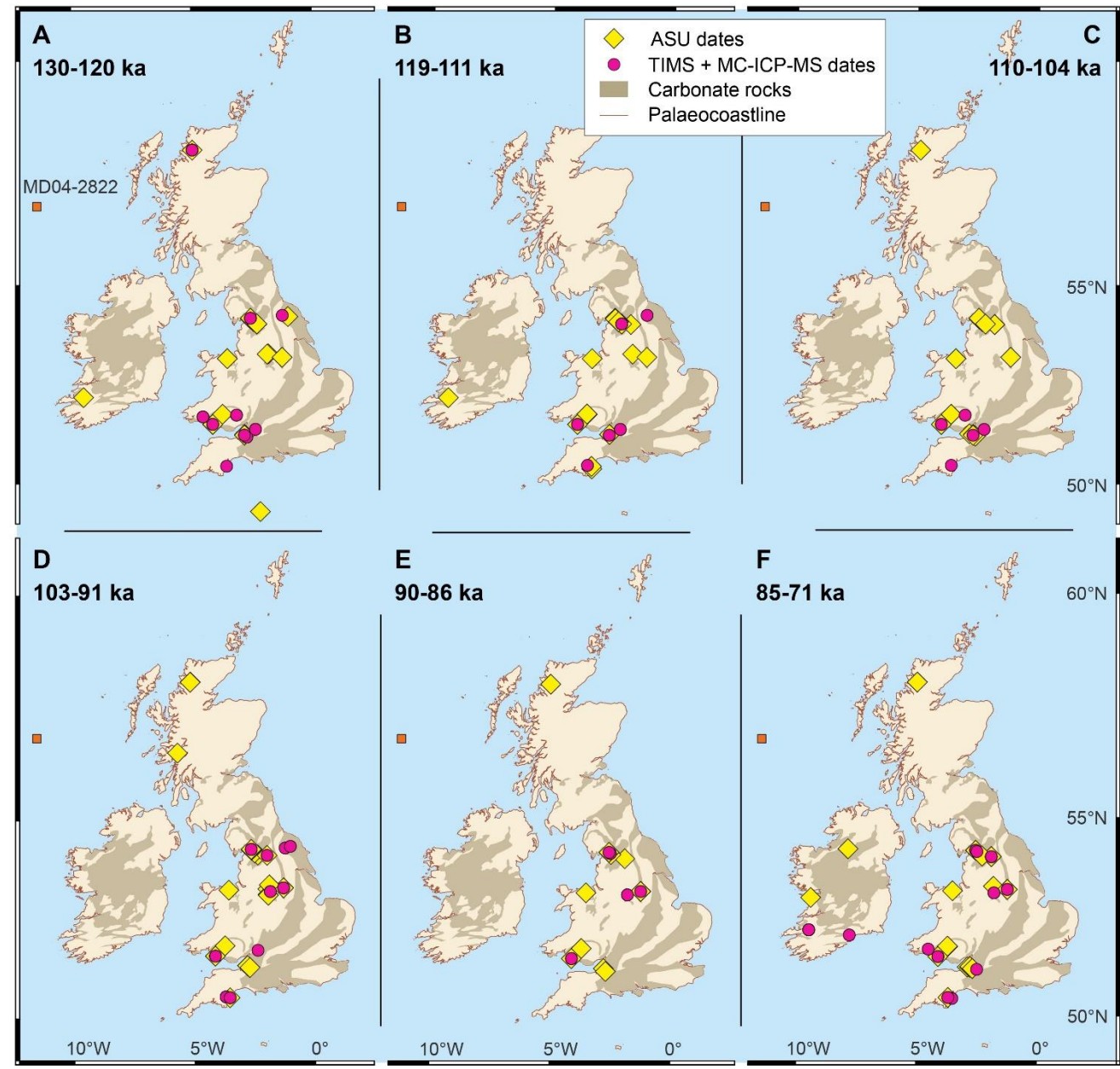

**Figure 8: Speleothem growth in the British Isles during selected time intervals from 130 to 71 ka, and location of site MD04-2822**
**(Hibbert et al., 2010). Distribution of carbonate rocks from World Karst Aquifer Mapping project (Chen et al., 2017).**

Between 130 and 120 ka, which covers interglacial MIS 5e, speleothem growth across England and northern Scotland indicates widespread favourable climatic conditions for speleothem deposition (i.e., absence of glacier ice or continuous permafrost). The absence of MS dates from Ireland can likely be ascribed to sampling bias (Fig. 8). After MIS5e, speleothem growth is restricted to England and Wales between 119 and 111 ka, with the northernmost dated speleothems located in

northern England (54°N; Fig. 8). The absence of speleothem growth in Scotland at this time may be indicative of cooling towards stadial MIS 5d as reflected in vegetation records from Europe (Helmens, 2014), but could also be the result of undersampling in this relatively karst-poor region.

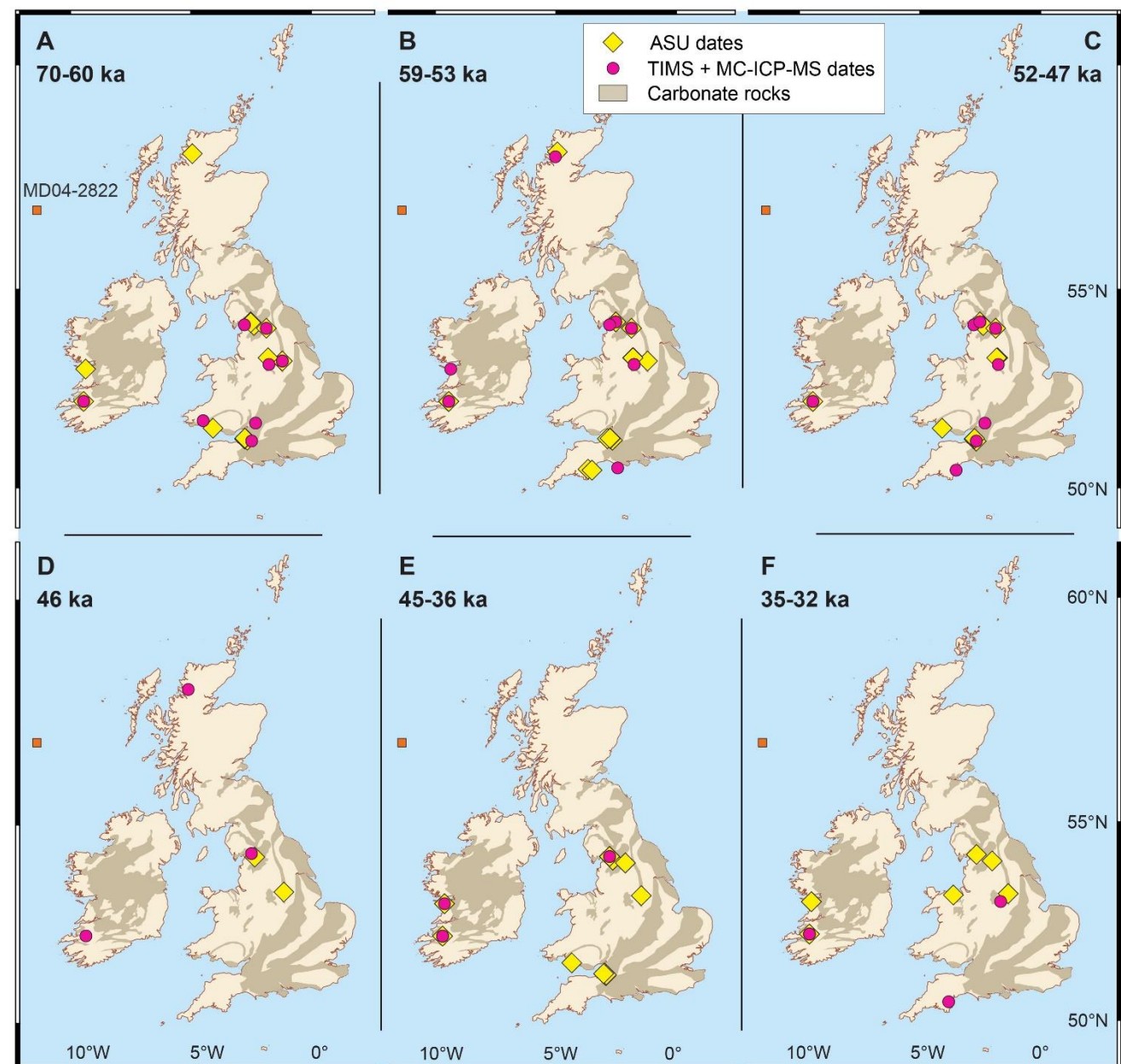

Figure 9: Speleothem growth in the British Isles during selected time intervals from 70 to 32 ka, and location of site MD04-2822 (Hibbert et al., 2010). Distribution of carbonate rocks from World Karst Aquifer Mapping project (Chen et al., 2017).

During MIS 5d, speleothem growth occurred sporadically from 110 to 104 ka in SW England based on the MS dataset. However, according to the ASU dataset speleothem growth was possible across Britain, with one date in northern Scotland at 105 ka (2σ error = ±3.7 ka), suggesting an extensive ice sheet did not establish in Scotland during MIS 5d (Fig. 8C). In the subsequent interstadial (MIS 5c, ~100 ka) and stadial (MIS 5b, ~95 ka), speleothem growth occurred in England and Wales

between 103 and 86 ka based on the MS dates (Fig. 8), and in Scotland at 100, 95 and 89-88 ka based on ASU dates, suggesting relatively mild climatic conditions (Fig. 8D,E). Between 85 and 71 ka (MIS 5a), speleothem growth is more abundant across England, Wales and SW Ireland, but some dates are recorded in Scotland (Fig. 8F). Overall, the presence of 22 ASU dates from NW Scotland between 130 and 71 ka suggests continued or at least repeated speleothem growth throughout MIS 5. The geographic difference between MS and ASU dates likely reflects the timing of speleothem studies (Lawson and Atkinson, 1995), rather than inherent differences in the techniques, but reassessment of ASU dates with MS techniques would allow confirmation of these dates and reduce associated age uncertainties.

Based on the speleothem data, an extensive ice sheet did not cover the British Isles between 70 and 60 ka, a conclusion supported by the reporting of oak pollen in speleothems from the Yorkshire Dales at this time (Caseldine et al., 2008) However, these pollen grains may have been reworked from older peat soils in the region deposited during the last interglacial. Our data suggest regional ice sheets may have been present in Scotland and northern Ireland (Fig. 9A) However, marine evidence suggests full glacial conditions across Europe and the North Atlantic (Helmens et al. 2014, and references therein) and an extensive grounded ice sheet across the North Sea, merging the British and Scandinavian ice sheets (Carr et al. 2006; Graham et al. 2011). Because the timing and extent of glaciations during MIS 4 remain poorly constrained by surface studies, the speleothems may provide the best direct evidence of the (lack of) ice cover at this time. This apparent conflict merits further investigation.

At the beginning of MIS 3 (c. 60 ka), ASU dates are recorded in Scotland between 59 and 53 ka, while MS dates continue to 53 and 46 ka  (Fig. 9; Lawson et al 2023). These two growth phases may be linked to Greenland Interstadials 14 and 12, highlighting a rapid response of a Scottish ice sheet margin to warming (Lawson et al., 2023), analogous to Crag Cave (Fankhauser et al., 2016).

The absence of Scottish speleothem dates after 46 ka (Fig. 9) is consistent with surface deposits that suggest the expansion of a major Scottish ice sheet, a precursor to the last BIIS, between 44 and 38 ka (Bradwell et al., 2021), However, surface deposits suggest a return to milder climatic conditions and Scotland-wide ice-free conditions by 38-37 ka (Whittington and Hall, 2002; Bradwell et al., 2021), which is not yet reflected in the speleothem growth data. Whilst organic sediments from the Isle of Lewis off NW Scotland suggest ice-free conditions until c. 32 ka (Whittington and Hall, 2002), Bradwell et al. (2021) suggest that ice masses probably existed in mainland Scotland from c. 35 ka onwards. This is supported by evidence of tundra and discontinuous permafrost in southern Scotland around 36 ka (Brown et al., 2007). Therefore, the absence of speleothem dates in NW Scotland since c. 36-35 ka likely reflects the expanding last glacial BIIS which was well established by 31 ka (Bradwell et al., 2021; Clark et al., 2022a).

## 5 Implications for glacial reconstructions

The presented compilation of speleothem dates demonstrates that speleothem growth in the British Isles can be used to reconstruct the presence and absence of ice sheets and continued permafrost for past glaciations. MS dates offer a more temporally constrained view of speleothem growth phases compared to ASU dates. But at present, undersampling in time and space using modern MS techniques is a key limitation for the spatial reconstruction in the British Isles, and this sampling bias likely causes apparent discontinuous growth and inconsistencies between the ASU and MS datasets. Discrepancies between ASU-dated speleothem growth phases and ice sheet limits likely result from the lower accuracy of this dating method compared to mass spectrometric dating techniques. However, the large number of ASU-dated samples available across the British Isles highlights the opportunity for re-analysis of archived material using more precise dating approaches.

Furthermore, the Crag Cave and Assynt datasets highlight the potential of speleothem U-series dating at ice-marginal sites to reconstruct millennial-scale stadials and interstadials for fine-tuning the BIIS reconstruction. With a higher sampling density of speleothem dates this capability can be extended to regional ice sheet growth and help constrain the dynamics of the BIIS. Considering the uncertainties inherent within both the speleothem and BIIS reconstruction will be key to synthesising surface and subsurface data.

In addition, determining the lead and lag times between the speleothem growth and ice advance/retreat at the surface can provide critical insights into the dynamics of permafrost formation/thaw, soil interactions, and hydroclimate. Re-evaluation of existing samples and expansion of the dataset within key regions are priorities for future research, with the reconstruction of glacial dynamics in the British Isles during MIS 4 and MIS 3 being a clear target. Another priority for further sampling is material older than the last interglacial, which is currently underrepresented in the dataset.

### Data availability

The database compiled and used in the study and the calculated Probability Density Function curves are available at 10.6084/m9.figshare.26084536.

### Competing interests

The contact author has declared that none of the authors has any competing interests.

## Acknowledgements

The work was made possible through the foundational speleothem research in Britain carried out by the late Mel Gascoyne, by Henry P. Schwarcz, Derek C. Ford, Andy Baker and others. We thank the Quaternary Research Association [Quaternary Research Award 2022, awarded to T.C.A. and T.J.L.] and the British Cave Research Association [Cave Science and Technology Research Fund, awarded to S.F.M.B.] for financial support for the U-Th dating of Scottish speleothem samples. We thank Hamish Cooper for sample preparation and U-Th dating at the University of Oxford.

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

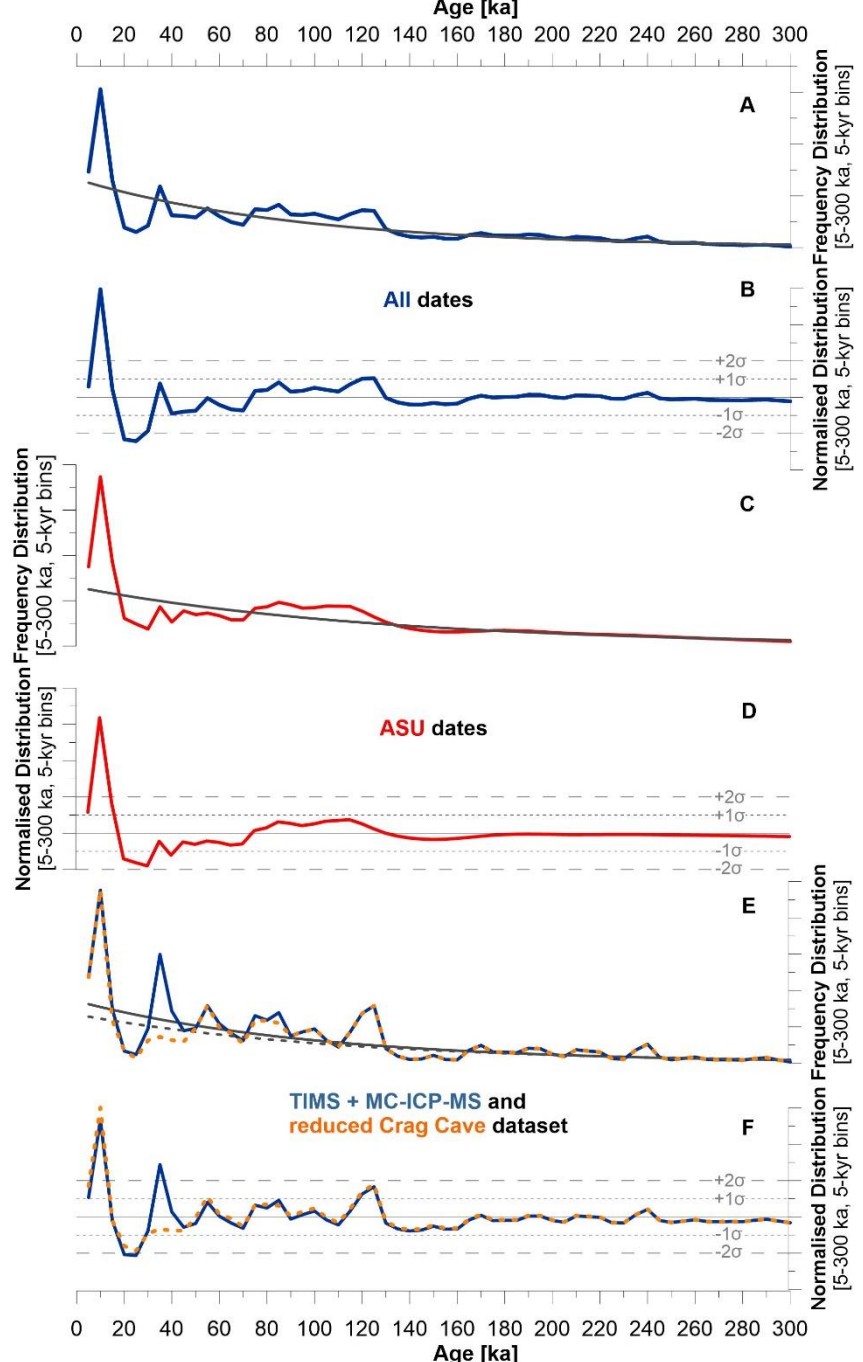

**Figure A1: Frequency distributions of speleothem growth between 300 and 5 ka based on the different datasets. (A, C, E) Probability density function with exponential trendline. (B, D, F) Frequency distribution following the subtraction of the exponential baseline and normalisation using z-score (see Methods).**

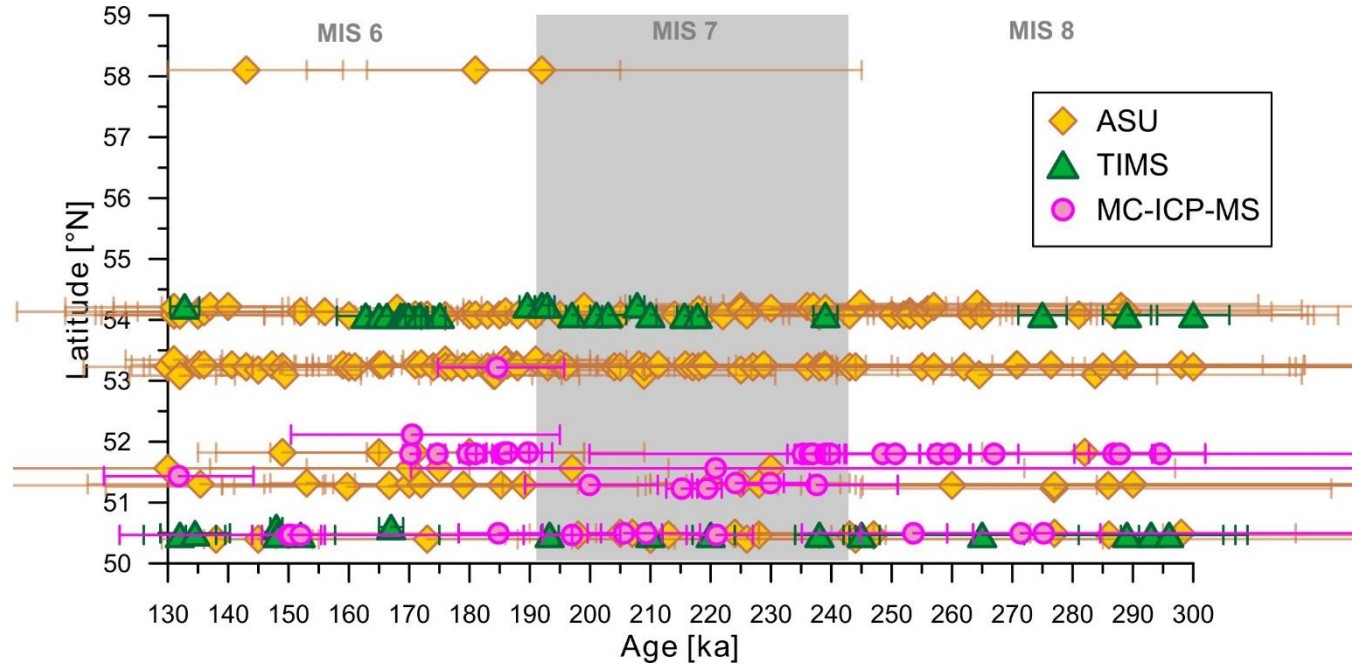


**Figure A2: Distribution of speleothem ages in the British Isles between 300 and 130 ka, covering Marine Isotope Stages MIS 8 to 6.**

**Appendix B: List of references used for compilation of dates.**

Channel Islands

Keen, D. H., Harmon, R. S., and Andrews, J. N.: U series and amino acid dates from Jersey, Nature, 289, 162–164,
https://doi.org/https://doi.org/10.1038/289162a0, 1981.

England

Atkinson, T. C., Harmon, R. S., Smart, P. L., and Waltham, A. C.: Palaeoclimatic and geomorphic implications of
230Th/234U dates on speleothems from Britain, Nature, 272, 24–28, https://doi.org/10.1038/272024a0, 1978.

Atkinson, T. C., Smart, P. L., and Andrews, J. N.: Uranium-series dating of speleothems from Mendip Caves. 1: Rhino Rift,
Charterhouse-on-Mendip, Proc. Univ. Bristol Spelaeol. Soc., 17, 55–69, 1984.

Atkinson, T. C., Lawson, T. J., Smart, P. L., Harmon, R. S., and Hess, J. W.: New data on speleothem deposition and
palaeoclimate in Britain over the last forty thousand years, J. Quat. Sci., 1, 67–72, https://doi.org/10.1002/jqs.3390010108,
1986.

Baker, A.: Speleothem growth rate and palaeoclimate., University of Bristol, 232 pp., 1993.

Baker, A., Smart, P. L., and Lawrence Edwards, R.: Paleoclimate implications of mass spectrometric dating of a British
flowstone, Geology, 23, 309, https://doi.org/10.1130/0091-7613(1995)023<0309:PIOMSD>2.3.CO;2, 1995.

Baker, A., Smart, P. L., and Edwards, R. L.: Mass spectrometric dating of flowstones from Stump Cross Caverns and Lancaster Hole, Yorkshire: palaeoclimate implications, J. Quat. Sci., 11, 107–114, https://doi.org/10.1002/(SICI)1099-1417(199603/04)11:2<107::AID-JQS236>3.0.CO;2-E, 1996.

Caseldine, C. J., McGarry, S. F., Baker, A., Hawkesworth, C., and Smart, P. L.: Late Quaternary speleothem pollen in the British Isles, J. Quat. Sci., 23, 193–200, https://doi.org/https://doi.org/10.1002/jqs.1121, 2008.

Daley, T. J., Thomas, E. R., Holmes, J. A., Street-Perrott, F. A., Chapman, M. R., Tindall, J. C., Valdes, P. J., Loader, N. J., Marshall, J. D., Wolff, E. W., Hopley, P. J., Atkinson, T., Barber, K. E., Fisher, E. H., Robertson, I., Hughes, P. D. M., and Roberts, C. N.: The 8200yr BP cold event in stable isotope records from the North Atlantic region, Glob. Planet. Change, 79, 288–302, https://doi.org/10.1016/j.gloplacha.2011.03.006, 2011.

Farrant, A. R., Noble, S. R., Barron, A. J. M., Self, C. A., and Grebby, S. R.: Speleothem U-series constraints on scarp retreat rates and landscape evolution: an example from the Severn valley and Cotswold Hills gull-caves, UK, J. Geol. Soc. London., 172, 63–76, https://doi.org/10.1144/jgs2014-028, 2015.

Ford, T. D.: The Evolution of the Castleton Cave Systems, Derbyshire, Cave Karst Sci., 13, 131–148, 1986.

Ford, T. D., Gascoyne, M., and Beck, J. S.: Speleothem dates and Pleistocene Chronology in the Peak District of Derbyshire, Cave Karst Sci., 10, 1983.

Gascoyne, M.: DOES THE PRESENCE OF STALAGMITES REALLY INDICATE WARM PERIODS? NEW EVIDENCE FROM YORKSHIRE AND CANADIAN CAVES, in: PROCEEDINGS OF THE 7th INTERNATIONAL SPELEOLOGICAL CONGRESS SHEFFIELD, ENGLAND SEPTEMBER, 1977, 208–210, 1977.

Gascoyne, M., Schwarcz, H. P., and Ford, D. C.: Uranium-series ages of speleothem from northwest England: correlation with Quaternary climate, Philos. Trans. R. Soc. London. B, Biol. Sci., 301, 143–164, https://doi.org/10.1098/rstb.1983.0024, 1983.

Gilmour, M., Currant, A., Jacobi, R., and Stringer, C.: Recent TIMS dating results from British Late Pleistocene vertebrate faunal localities: context and interpretation, J. Quat. Sci., 22, 793–800, https://doi.org/10.1002/jqs.1112, 2007.

Gordon, D.: The Pleistocene of the Mendip region : Aspects of the absolute dated, faunal and sediment records., University of Bristol, 370 pp., 1987.

Green, C. P., Coope, G. R., Currant, A. P., Holyoak, D. T., Ivanovich, M., Jones, R. L., Keen, D. H., McGregor, D. F. M., and Robinson, J. E.: Evidence of two temperate episodes in late Pleistocene deposits at Marsworth, UK, Nature, 309, 778–781, https://doi.org/10.1038/309778a0, 1984.

Gunn, J., Fairchild, I., Moseley, G., Töchterle, P., Ashley, K. E., Hellstrom, J., and Edwards, R. L.: Palaeoenvironments in the central White Peak District ( Derbyshire , UK ): evidence from Water Icicle Close Cavern, Cave Karst Sci., 47, 153–168, 2020.

Hodge, E., Hoffmann, D. L., Richards, D. A., and Smart, P. L.: Uranium-series ages for speleothem and tufa deposits associated with Quaternary mammalian fossil evidence in England and Wales, Proc. Univ. Bristol Spelaeol. Soc., 27, 73–80, 2016.

Jacobi, R. M., Rowe, P. J., Gilmour, M. A., Grün, R., and Atkinson, T. C.: Radiometric dating of the Middle Palaeolithic tool industry and associated fauna of Pin Hole Cave, Creswell Crags, England, J. Quat. Sci., 13, 29–42, https://doi.org/10.1002/(SICI)1099-1417(199801/02)13:1<29::AID-JQS346>3.0.CO;2-6, 1998.

Latham, A. P., Schwarcz, H. P., Ford, D. C., and Pearce, G. W.: Palaeomagnetism of stalagmite deposits, Nature, 280, 283–385, https://doi.org/https://doi.org/10.1038/280383a0, 1979.

Lundberg, J. and McFarlane, D. A.: Pleistocene depositional history in a periglacial terrane: A 500 k.y. record from Kents Cavern, Devon, United Kingdom, Geosphere, 3, 199, https://doi.org/10.1130/GES00085.1, 2007.

Lundberg, J., Simons, J., and McFarlane, D. A.: A Pleistocene chronology for the fauna and artefacts of Cow Cave, Devon, 645 UK., Cave Karst Sci., 34, 101–104, 2007.

Lundberg, J., Lord, T. C., and Murphy, P. J.: Thermal ionization mass spectrometer U-Th dates on Pleistocene speleothems from Victoria Cave, North Yorkshire, UK: Implications for paleoenvironment and stratigraphy over multiple glacial cycles, Geosphere, 6, 379–395, https://doi.org/10.1130/GES00540.1, 2010.

Lundberg, J., Lord, T. C., and Murphy, P. J.: New U-series dates from Stump Cross Caverns, Yorkshire, UK, and constraints 650 on the age of the Banwell Bone Cave Mammal Assemblage Zone, Proc. Geol. Assoc., 131, 639–651, https://doi.org/10.1016/j.pgeola.2020.04.004, 2020.

McFarlane, D. A., Lundberg, J., and Ford, D. C.: The Age of the Woolly Rhino from Dream Cave, Derbyshire, UK, Cave Karst Sci., 27, 25–28, 2000.

McFarlane, D. A., Lundberg, J., and Cordingley, J.: A brief history of stalagmite growth measurements at Ingleborough 655 Cave, Yorkshire, United Kingdom., Cave Karst Sci., 31, 113–118, 2004.

McFarlane, D. A., Lundberg, J., and Neff, H.: A Speleothem Record of Early British and Roman Mining at Charterhouse, Mendip, England, Archaeometry, 56, 431–443, https://doi.org/10.1111/arcm.12025, 2014.

Mullan, G. J. and Moody, A. A. D.: An account and survey of Great Oone's Hole, Cheddar Gorge, Somerset, Proc. Univ. Bristol Spelaeol. Soc., 26, 117–130, 2014.

Murphy, P. J. and Graham, N.: Uranium series dates from mass movement caves on the Isle of Portland, Dorset, UK, Proc. Univ. Bristol Spelaeol. Soc., 25, 113–115, 2010.

Murphy, P. J. and Lowe, D. J.: A uranium-series date from a karst cave on the North York Moors, Yorkshire, UK, Cave Karst Sci., 35, 105–106, 2008.

Murphy, P. J. and Lundberg, J.: Uranium series dates from the windy pits of the North York Moors, United Kingdom: 665 implications for late Quaternary ice cover and timing of speleogenesis, Earth Surf. Process. Landforms, 34, 305–313, https://doi.org/10.1002/esp.1752, 2009.

Murphy, P. J. and Moseley, G. E.: A uranium-series date from Wall End Cave, Silverdale, northwest England: its palaeoclimatic and archaeological significance, Cave Karst Sci., 47, 138–140, 2020.

Murphy, P. J., Smallshire, R., and Midgley, C.: The sediments of Illusion Pot, Kingsdale, UK: Evidence for sub-glacial 670 utilisation of a karst conduit in the Yorkshire Dales?, Cave Karst Sci., 28, 29–34, 2001.

Murphy, P. J., Lundberg, J., and Cordingley, J.: A uranium-series date from Keld Head Kingsdale, North Yorkshire, UK., Cave Karst Sci., 31, 77–78, 2004.

Murphy, P. J., Cordingley, J., and Waltham, T.: New uranium-series dates from Keld Head, Kingsdale, North Yorkshire, UK., Cave Karst Sci., 35, 111–114, 2008.

Murphy, P. J., Hodgson, D., Richards, D. A., and Standish, C. D.: Boreham Cave, Littondale, North Yorkshire, UK: some geomorphological observations., Cave Karst Sci., 40, 109–113, 2013.

Murphy, P. J., Moseley, G. E., Moseley, M., and Edwards, R. L.: Preliminary uranium-series ages and stable-isotopes from Fairy Hole, Warton Crag, Lancashire, UK:implications for speleogenesis and palaeoclimate, Cave Karst Sci., 43, 103–106, 2016.

Pike, A. W. G., Gilmour, M., Pettitt, P., Jacobi, R., Ripoll, S., Bahn, P., and Muñoz, F.: Verification of the age of the Palaeolithic cave art at Creswell Crags, UK, J. Archaeol. Sci., 32, 1649–1655, https://doi.org/10.1016/j.jas.2005.05.002, 2005.

Proctor, C. J.: A British Pleistocene chronology based on uranium series and electron spin resonance dating of speleothem., University of Bristol, 353 pp., 1994.

Proctor, C. J. and Smart, P. L.: A dated cave sediment record of Pleistocene transgressions on Berry Head, Southwest England, J. Quat. Sci., 6, 233–244, https://doi.org/10.1002/jqs.3390060306, 1991.

Proctor, C. J., Berridge, P., Bishop, M., Richards, D. A., and Smart, P. L.: Age of Middle Pleistocene fauna and Lower Palaeolithic industries from Kent's Cavern, Devon, Quat. Sci. Rev., 24, 1243–1252, https://doi.org/10.1016/j.quascirev.2004.07.022, 2005.

Roberts, M. S., Smart, P. L., Hawkesworth, C. J., Perkins, W. T., and Pearce, N. J. G.: Trace element variations in coeval Holocene speleothems from GB Cave, southwest England, The Holocene, 9, 707–713, https://doi.org/10.1191/095968399672615014, 1999.

Rowe, P. J.: Uranium-series dating of cave sites in the English Midlands, University of East Anglia, 1986.

Rowe, P. J., Atkinson, T. C., and Jenkinson, R. D. S.: Uranium-Series Dating of Cave Deposits at Creswell Crags Gorge, 695 England., Cave Karst Sci., 16, 3–16, 1989.

Simmonds, V.: Evidence for Pleistocene frost and ice damage of speleothems in Hallowe'en Rift, Mendip Hills, Somerset, UK, Cave Karst Sci., 46, 74–78, 2019.

Sutcliffe, A. J., Lord, T. C., Harmon, R. S., Ivanovich, M., Rae, A., and Hess, J. W.: Wolverine in Northern England at About 83,000 yr B.P.: Faunal Evidence for Climatic Change during Isotope Stage 5, Quat. Res., 24, 73–86, 700 https://doi.org/10.1016/0033-5894(85)90084-5, 1985.

Waltham, T., Murphy, P., and Batty, A.: Kingsdale: the evolution of a Yorkshire dale, Proc. Yorksh. Geol. Soc., 58, 95–105, https://doi.org/10.1144/pygs.58.1.277, 2010.

Ireland

Fankhauser, A., McDermott, F., and Fleitmann, D.: Episodic speleothem deposition tracks the terrestrial impact of millennial-scale last glacial climate variability in SW Ireland, Quat. Sci. Rev., 152, 104–117, https://doi.org/10.1016/j.quascirev.2016.09.019, 2016.

Gunn, J., Rushworth, G., Hunt, C., and Lauritzen, S.-E.: Clastic sediments in Crag Cave, in: Guidebook for KR6 field trip, 22nd June 2011, edited by: Gunn, J., 10–13, 2011.

Lundberg, J. and Drew, D.: The antiquity of Aillwee Cave calcite deposits, Burren District, Ireland., Cave Karst Sci., 33, 9–10, 2006.

McDermott, F. and Fankhauser, A.: Episodic speleothem deposition in Ireland during the late Quaternary; implications for Greenland ice core chronology and British-Irish Ice Sheet dynamics, in: EGU General Assembly, 2016.

McDermott, F., Frisia, S., Huang, Y., Longinelli, A., Spiro, B., Heaton, T. H. E., Hawkesworth, C. J., Borsato, A., Keppens,
E., Fairchild, I. J., Van der Borg, K., Verheyden, S., and Selmo, E.: Holocene climate variability in Europe: Evidence from δ18O, textural and extension-rate variations in three speleothems, Quat. Sci. Rev., 18, 1021–1038, https://doi.org/10.1016/S0277-3791(98)00107-3, 1999.

McDermott, F., Mattey, D. P., and Hawkesworth, C.: Centennial-Scale Holocene Climate Variability Revealed by a High-Resolution Speleothem δ 18 O Record from SW Ireland, Science (80-. )., 294, 1328–1331,
https://doi.org/10.1126/science.1063678, 2001.

Simms, M. J.: Further insights into the Irish Quaternary from two other sites; Poulsallagh and Pol an Ionain, in: The Burren, Co. Clare. Irish Quaternary Association Field Guide no. 33., edited by: Nolan, J. and Randolph, C., Irish Quaternary Association, Dublin, 205–236, 2016.

Simms, M. J. and Coxon, P.: Glacial and karst landscapes of the Gort lowlands and Burren, Quat. Cent. West. Irel. F. Guid.,
39–63, 2005.

Swabey, S. E. J.: Rates of natural climate change: a study of speleothems, The Open University, https://doi.org/https://doi.org/10.21954/ou.ro.0000d519, 1996.

Vesely, M. M.: Annual Speleothern Laminae: Indicators of Paleoclimate in Ireland, Carleton University, 108 pp., 2000.

Northern Ireland

Gunn, J.: Marble Arch Caves Global Geopark (MACGG) County Fermanagh, UK & County Cavan, Republic of Ireland, 13 pp., 2011.

Scotland

Atkinson, T. C., Lawson, T. J., Smart, P. L., Harmon, R. S., and Hess, J. W.: New data on speleothem deposition and palaeoclimate in Britain over the last forty thousand years, J. Quat. Sci., 1, 67–72, https://doi.org/10.1002/jqs.3390010108, 1986.

Baker, A., Smart, P. L., and Ford, D. C.: Northwest European palaeoclimate as indicated by growth frequency variations of secondary calcite deposits, Palaeogeogr. Palaeoclimatol. Palaeoecol., 100, 291–301, https://doi.org/10.1016/0031-0182(93)90059-R, 1993.

Baker, A., Smart, P. L., Barnes, W. L., Edwards, R. L., and Farrant, A.: The Hekla 3 volcanic eruption recorded in a Scottish speleothem?, The Holocene, 5, 336–342, https://doi.org/10.1177/095968369500500309, 1995.

Blyth, A. J., Baker, A., Thomas, L. E., and Van Calsteren, P.: A 2000-year lipid biomarker record preserved in a stalagmite from north-west Scotland, J. Quat. Sci., 26, 326–334, https://doi.org/10.1002/jqs.1457, 2011.

Farrant, A. R., Smith, C. J. M., Noble, S. R., Simmas, M. J., and Richards, D. A.: Speleogenetic evidence from Ogof Draenen for a pre-Devensian glaciation in the Brecon Beacons, South Wales, UK, J. Quat. Sci., 29, 815–826, https://doi.org/10.1002/jqs.2751, 2014.

Genty, D., Baker, A., Massault, M., Proctor, C., Gilmour, M., Pons-Branchu, E., and Hamelin, B.: Dead carbon in stalagmites: carbonate bedrock paleodissolution vs. ageing of soil organic matter. Implications for 13 C variations in speleothems, Geochim. Cosmochim. Acta, 65, 3443–3457, https://doi.org/10.1016/S0016-7037(01)00697-4, 2001.

Gilmour, M., Currant, A., Jacobi, R., and Stringer, C.: Recent TIMS dating results from British Late Pleistocene vertebrate faunal localities: context and interpretation, J. Quat. Sci., 22, 793–800, https://doi.org/10.1002/jqs.1112, 2007.

Gray, J. M. and Ivanovich, M.: Age of the main rock platform, western Scotland, Palaeogeogr. Palaeoclimatol. Palaeoecol., 68, 337–345, https://doi.org/10.1016/0031-0182(88)90050-8, 1988.

Hodge, E., Hoffmann, D. L., Richards, D. A., and Smart, P. L.: Uranium-series ages for speleothem and tufa deposits associated with Quaternary mammalian fossil evidence in England and Wales, Proc. Univ. Bristol Spelaeol. Soc., 27, 73–80, 2016.

Lawson, T. J. and Atkinson, T. C.: Quarternary Chronology, in: The Quaternary of Assynt and Coigach: field guide, edited by: Lawson, T. J., Quaternary Research Association, 12–18, 1995.

Nash, G. H., van Calsteren, P., Thomas, L., and Simms, M. J.: A discovery of possible Upper Palaeolithic parietal art in Cathole Cave, Gower peninsula, South Wales, Proc. Univ. Bristol Spelaeol. Soc., 25, 327–336, 2012.

Proctor, C. J.: A British Pleistocene chronology based on uranium series and electron spin resonance dating of speleothem., University of Bristol, 353 pp., 1994.

Proctor, C. J., Baker, A., Barnes, W. L., and Gilmour, M. A.: A thousand year speleothem proxy record of North Atlantic climate from Scotland, Clim. Dyn., 16, 815–820, https://doi.org/10.1007/s003820000077, 2000.

Sissons, J. B.: The So-Called High "Interglacial" Rock Shoreline of Western Scotland, Trans. Inst. Br. Geogr., 7, 205, https://doi.org/10.2307/622222, 1982.

Stringer, C. B., Currant, A. P., Schwarcz, H. P., and Collcutt, S. N.: Age of Pleistocene faunas from Bacon Hole, Wales, Nature, 320, 59–62, https://doi.org/10.1038/320059a0, 1986.

Wales

Debenham, N. C., Aitken, M. J., Walton, A. J., and Winter, M.: Thermoluminescence and uranium series dating of stalagmitic calcite, in: Studies of Pontnewydd Cave, Wales, edited by: Green, H. S., National Museum of Wales, Cardiff, 100–105, 1984.

Farrant, A. R., Smith, C. J. M., Noble, S. R., Simmas, M. J., and Richards, D. A.: Speleogenetic evidence from Ogof Draenen for a pre-Devensian glaciation in the Brecon Beacons, South Wales, UK, J. Quat. Sci., 29, 815–826, https://doi.org/10.1002/jqs.2751, 2014.

Gordon, D.: The Pleistocene of the Mendip region : Aspects of the absolute dated, faunal and sediment records., University of Bristol, 370 pp., 1987.

Hodge, E., Hoffmann, D. L., Richards, D. A., and Smart, P. L.: Uranium-series ages for speleothem and tufa deposits associated with Quaternary mammalian fossil evidence in England and Wales, Proc. Univ. Bristol Spelaeol. Soc., 27, 73–80, 2016.

Ivanovich, M., Rae, A. M. B., and Wilkins, M. A.: Brief report on dating the in situ stalagmitic floor found in the East Passage in 1982, in: Studies of Pontnewydd Cave, Wales, edited by: Green, H. S., National Museum of Wales, Cardiff, 98–785   99, 1984.

Nash, G. H., van Calsteren, P., Thomas, L., and Simms, M. J.: A discovery of possible Upper Palaeolithic parietal art in Cathole Cave, Gower peninsula, South Wales, Proc. Univ. Bristol Spelaeol. Soc., 25, 327–336, 2012.

Proctor, C. J.: A British Pleistocene chronology based on uranium series and electron spin resonance dating of speleothem., University of Bristol, 353 pp., 1994.

Schwarcz, H. P.: Uranium-series dating and stable isotope analyses of calcite deposits, in: Studies of Pontnewydd Cave, Wales, edited by: Green, H. S., National Museum of Wales, Cardiff, 88–97, 1984.

Stringer, C. B., Currant, A. P., Schwarcz, H. P., and Collcutt, S. N.: Age of Pleistocene faunas from Bacon Hole, Wales, Nature, 320, 59–62, https://doi.org/10.1038/320059a0, 1986.
