# Peer review of "Spatio-temporal dynamics of speleothem growth and glaciation in the British Isles"

_Climate of the Past, 2024_

## Author Comment (AC1)

**General Comment**

The authors report an impressive new synthesis of speleothem U-Th ages from the British Isles, and use this to focus on a comparison between last glacial ice sheet location and the speleothem growth. It is a very valuable contribution to the literature.

*We appreciate the suggestions of the reviewer and have used their constructive comments to improve our work. Please find our responses to the comments in the attached PDF, highlighted in blue. For our responses attached we use line numbers pertaining to the updated manuscript.*

**Specific Comments**

My questions to the authors largely relate to asking for more methodological detail so that the reader can reproduce the analyses and assess the certainty to which a speleothem U-Th age can be correctly attributed to the bins in section 4.2 'Spatial Distribution of Speleothem Growth'.

Line 98. Should the published sources of the compiled ages be cited somewhere e.g. a summary table by region? I see that the Figshare file does have complete or partial references, but that is separate from this document.

*We agree such table would be useful for future researchers. We now include a full reference list containing details of all studies cited in Data File 1 in the Appendix to this manuscript (Appendix B, Lines 587-770).*

On lines 103-109. The authors state that they have chosen to use the published corrected U-Th ages, with justifications given. Some dates were rejected (line 106). So that future researchers could reproduce and expand on this work, please specify the criterion used for rejection of ages.

*Our criteria were very simple: We remove any samples without dates, or with infinite ages, or infinite age uncertainties (or combinations of these parameters). We have clarified this in the updated manuscript (L106-107):*

*"When published, the age corrected for initial $^{230}$Th was used, with any samples without dates, with infinite dates or infinite uncertainties removed from the analysis."*

On line 107 it is stated that where uncertainties were not provided, the average was used (depending on analysis method). Was the age also taken into consideration e.g. as written, does this likely underestimate the uncertainty for older samples, and overestimate it for younger samples, and underestimate the uncertainty for samples that are detritally contaminated. Does the inclusion of these samples have an undue influence on the overall result?

*The average uncertainty values used are percentages, and so increase or decrease relatively with the absolute age of the sample. For example, an ASU date of 50 ka would be assumed to have a ±8.8 kyr uncertainty (using the average ASU uncertainty of 17.65% of the absolute age). For another ASU date, but with an age of 250 ka, the uncertainty thus would be ±44.1 kyr.*

*As such, we believe there is little chance of the inclusion of these samples impacting the overall reconstruction.*

Lines 123-150. Could the authors provide more detail on the pdf method, noting that it is 'following Scroxton et al (2016)' but that publication might not be accessible to all. In particular, could more information be provided on the calculation of the z-scores shown in Figure 2. Did

the authors generate '10,000 synthetic ages .... calculated from the same exponential relationship to determine the predicted variation through time caused by chance' (quoted from Scroxton et al 2016).

*We now include more information on the specific approach used here (L148-154):*

*"For this, we fit an exponential function to the dataset of age versus frequency ($y = 13.388e^{-0.009x}$). Then, for each point in time we subtracted the expected value (i.e if the function fitted the data perfectly) from the observed value, thereby removing the underlying 'natural attrition' trend that reduces the height (depth) of peaks (troughs) with time to allow for better comparison of relative peak heights. These values are then converted to standard scores (z-scores) to allow for the variability to be more easily visualised."*

*However, we did not generate 10,000 synthetic ages. As removal of an exponential function from the dataset is a simple subtraction of the expected value if the function represents variability of 100% of the data, we do not feel the need to consider synthetic ages here.*

*Furthermore, as the Scroxton et al. paper demonstrated, there is no evidence that the removal of the exponential function resulted in a random dataset.*

Section 4.2.1. Would the authors consider including some expert opinion on the quality of some of the critical sites that are considered in this section. For example, Crag Cave appears to be very important when comparing the timing of the ice sheet in the west of Ireland, but how reliable are those dates? I went to look at the supplemental datafile to see if one estimate of reliability, the $^{230}$Th/$^{232}$Th in the speleothem, had been compiled, and that is not the case. However, I had access to the original publication (Fankhauser et al 2016, which might be paywalled for others) and could check that these speleothems had precise dates with very low detrital Th contamination. I would suggest that this would be useful information to convey in the text, and similarly for any other sites and samples that are critical to the interpretations made in section 4.

*This section does indeed lean heavily on the data from Crag Cave. As mentioned by the reviewer, the dates from this cave are very high quality and we consider this site suitable for establishing the chronology of ice sheet dynamics, because detrital contamination is low, U content is high, and so the ages are considered highly reliable. As such, we now highlight this fact in the text (L277).*

*No widely agreed, objective framework for judging where a line can be drawn in terms of "quality" is available in the literature as far as we are aware. While we could create such a framework ourselves, that would represent a significant expansion of the scope of this study. Consequently, as discussed in our reply to Reviewer 2, for the purpose of this paper we presume that data quality was audited at the point of peer review of original studies. The logical next step for the research following this paper will require a redating programme which includes all the existing data being brought into a common calibration framework as recommended by Reviewer 2. This is the logical point to consider establishing a "quality" framework to determine which dates are taken on, and which need to be re-done.*

Lines 395-406. With apologies for the self-citation, a comment that Caseldine et al. (2008) report oak pollen in speleothems from the Yorkshire Dales well into this time period, which would agree with this interpretation.

*We appreciate the suggestion and include the reference in the updated work (L405-406):*

*"Based on the speleothem data, an extensive ice sheet did not cover the British Isles between 70 and 60 ka. This is a conclusion supported by the reporting of oak pollen in speleothems from the Yorkshire Dales at this time (Caseldine et al., 2008). However, these pollen grains may have been reworked from older peat soils in the region deposited during the last interglacial. Our data suggest regional ice sheets may have been present in Scotland and northern Ireland (Fig. 9A)."*

References

Caseldine, C.J., McGarry, S.F., Baker, A., Hawkesworth, C. and Smart, P.L.: Late Quaternary speleothem pollen in the British Isles. *Journal of Quaternary Science*, 23, 193-200, https://doi.org/10.1002/jqs.1121, 2008.

Fankhauser, A., McDermott, F., and Fleitmann, D.: Episodic speleothem deposition tracks the terrestrial impact of millennial-scale last glacial climate variability in SW Ireland, Quat. Sci. Rev., 152, 104–117, https://doi.org/10.1016/j.quascirev.2016.09.019, 2016.

Scroxton, N., Gagan, M. K., Dunbar, G. B., Ayliffe, L. K., Hantoro, W. S., Shen, C. C., Hellstrom, J. C., Zhao, J. X., Cheng, H., Edwards, R. L., Sun, H., and Rifai, H.: Natural attrition and growth frequency variations of stalagmites in southwest Sulawesi over the past 530,000 years, Palaeogeogr. Palaeoclimatol. Palaeoecol., 441, 823–833, https://doi.org/10.1016/j.palaeo.2015.10.030, 2016.

---

## Author Comment (AC2)

My sincere apologies to the authors and editor for the time taken for this review.

This manuscript is an important contribution in its demonstration of the potential for constraining the extent and distribution of ice cover over time where cave records are available. It is somewhat hamstrung by the quality of a lot of the available data (Alpha spectrometric age determinations, predominantly conducted before the development of mass spectrometric dating in the late 1980s) but clearly illustrates the potential for a large project to address this in the UK. The study also clearly points to the potential for constraining timing and extent of older glacial advances for which the surface evidence no longer exists, although this would require a great deal of U-Th dating to locate older speleothem material.

*We appreciate the reviewer's positive assessment of our work.*

The authors have chosen to use U-Th ages as originally published, which allows use of a larger dataset where the isotopic data required to recalculate the ages using current techniques were not always provided. I would argue that it would have been preferable to recalculate ages for which sufficient isotopic data were available and discard those where they were not (mostly ASU ages I would expect), giving a cleaner less noisy record at the cost of reduced data density - but I can accept this is a matter of opinion and going back to do that at this point would be an enormous task beyond the scope of this study

*Whilst we also agree recalculation of ages may have been an option for this project, it is indeed beyond the scope of this work. The collation of published age information alone was a significant commitment and at this stage, the work was intended to represent a full review of all data available. Going forward, we intend to recalculate ages for those which sufficient data is available, alongside the collection of new data from locations where it's currently unavailable. This is, as the Reviewer suggests, an enormous task!*

I recommend publication. I have some comments below for the authors consideration, they mostly affect wording with the exception of the error noted below in Fig 6 and the raw data table which should be corrected.

line 104: I understand the difficulty of compiling data from sources which did not always include the required information to recalculate published ages but it should be noted that Gaussian distribution of the calculated age is not the only consideration here. Our knowledge of the relevant constants has improved significantly since some of these were published, as have practices for calculating corrected ages (for instance many publications around the turn of the century incorrectly calculated the uncertainties of their corrected ages due to either a bug or common misconfiguration of widely used software at that time). Without this information, some of these dates might be best viewed as qualitative. I don't think this is a show stopper, but it is perhaps an unnecessary source of noise affecting this study

*We acknowledge these issues are indeed potential sources of noise in the dataset, but as mentioned above, the recalculation is beyond the scope of this work (and in a number of cases impossible because of lack of data). We aim to perform rigorous reanalysis in future work.*

*The fact we are able to see the impact of ice sheet growth and retreat in such a potentially noisy dataset is clear grounds for optimism of what we will be able to resolve, and how far back we will be able to resolve it in a future study. It is likely the evaluation we present here of the potential for this approach to reveal new details of ice sheet dynamics is more pessimistic than it needs to be.*

*We now add a sentence to the manuscript to acknowledge this point (L105-106):*

*"We are also aware that the calculation and correction of ages has changed in the past 30 years, but recalculation for many ages is not possible, and we prefer to maximise the data density for this work"*

line 108: Uncertainties vary a lot with age, and older ages will bias the average age uncertainty determined here. Might be better practice to use median although with low number of old ages might not make much difference. Is the average uncertainty determined using the age-filtered dataset?

*Whilst we agree a bias may exist in this approach, we retain the use of the average as it allows us to consider error in a more conservative way as the average is higher than the median. Future work (see comments above) will consider this issue, and hopefully allow us to assess the magnitude of such potential bias. This again is grounds for optimism for what can be achieved by continuation of this research.*

115: I don't follow the reasoning for excluding young ages on this basis, needs to be more clearly explained. I can understand they might flood the record otherwise but I wouldn't call that a sampling artefact.

*We apologise for any confusion here, but dates <5 ka were removed as we expect, due to conservation and scientific research question reasons, that fewer samples of this age would be collected. To remove the impact of younger (but older than 5 ka) dates 'flooding' the dataset, we perform the exponential function removal as detailed in the methods. To clarify this point we now include the following in the text (L117-119):*

*"...active speleothems were generally not collected for conservation reasons and sampling is often done with the intention of collecting older material, resulting in a reduced number of samples available towards present (Gordon et al., 1989)."*

118: The question of how to address multiple dates on a single speleothem when compiling them into a histogram or PDF is always difficult. This approach intrinsically assumes random sampling, which is possible to do after a fashion if you set out into the field to sample randomly specifically for this purpose. But with a literature compilation approach (as most studies including this one have used) the difficulty is that the compiled data were mostly sampled anything but randomly, for completely different purposes. Where you have multiple dates available from a single speleothem the most appropriate approach might be to randomly choose one date and move on, but that is not really feasible, not least in that the reader must be convinced your choice was truely random. If you always take the oldest date your compilation is biased towards growth initiation times, if you take the youngest it is growth cessation times. I don't think there is an easy answer to this, but I would be wary of biasing the compiled record by including more than two or three dates from any individual speleothem, unless it has clear major growth hiatuses then maybe treat each interval separately. Certainly, reducing the Crags Cave dataset at least as much as done here is justified.

*We agree with the reviewer - this is a complex issue. To add to that picture, should two growth internals separated by a hiatus or drip axis shift be considered a single sample (with one random date taken) or multiple samples (with dates taken from each growth sequence)? As with the issue of date "quality", we lack an internationally agreed framework to work from. The Crag Cave dataset constitutes a (positive) outlier among the other cave sites in that it has a very tightly*

*fixed chronology. For other samples with fewer dates, randomization is further complicated by numerous distinct changes in the petrographic appearance of the dated samples, which mostly represent hiatuses of variable length. Without access to the original samples, determining how many dates to extract becomes very subjective. For older samples, petrographic or drip axis changes cannot reliably be assigned to be hiatuses due to larger uncertainties which may even be larger than the growth interruption itself, complicating subjective judgement further. We tried to find a middle ground without using our subjective judgment more than was strictly necessary.*

Page 5: I agree, 5 kyr binning seems like a reasonable compromise here. The only alternative is bin width as a non-linear function of age and I don't think anyone has gone there yet.

*We agree, determining bin size as a function of age would be one possibility but we prefer at this stage to stay with a simpler fixed bin size.*

Page 5: The exponential normalisation is glossed over here. What was the exponential decay coefficient used in each case (for instance, expressed in terms of probability of speleothem removal per kyr) and how was this fitted to the data? Figure A1 shows this graphically but doesn't explain how the fit is weighted to the data. The rate of decay used here would appear to have an impact on which of the normalised peaks are found to be significant at the next step.

*As discussed in our response to reviewer 1, we now include more detail in the updated manuscript on this approach (L148-154):*

*"For this, we fit an exponential function to the dataset of age versus frequency ($y = 13.388e^{-0.009x}$). Then, for each point in time we subtracted the expected value (i.e if the function fitted the data perfectly) from the observed value, thereby removing the underlying 'natural attrition' trend that reduces the height (depth) of peaks (troughs) with time to allow for better comparison of relative peak heights. These values are then converted to standard scores (z-scores) to allow for the variability to be more easily visualised."*

Line 151: I certainly agree with this. Constant removal from consideration by erosion/burial might not hold up very well at thousands of years timescale but should be a reasonable model over hundreds of thousands of years, as seen in multiple previous studies from around the world.

*Thanks, we agree.*

Figs. 2, 3 and A2, and results section: These make a pretty convincing case for disregarding or at least minimising the use of the ASU data. It was a revolutionary breakthrough for its time and it is important to have compiled all that was available, but mixing these with mass spectrometric U-Th data degrades the overall quality of the dataset. A similar argument applies to figure 6.

*We fully agree, but it is important to recognise the hard work and diligent science which is represented by those ASU dates – they are the foundation our work is built on. This work has demonstrated that ASU dates have a much lower accuracy than those from TIMS or MC-ICP-MS, and that it is time for a renewed attempt to reconstruct permafrost and ice dynamics with modern techniques. As such, we retain the presentation of ASU dates in these figures.*

196: I am always a little uneasy about hanging anything important from a single U-Th date. A cluster yes, even a pair, but there are just too many ways a single U-Th date can be wrong (due either to the sample itself, or potentially to the analytical process especially for ASU or TIMS).

*We agree, and have added the words 'accurate and reliable' here to qualify our statement.*

In general the discussion section is very good, with adequate recognition of the limitations of the ASU data combined with advocacy for extending the mass spectrometric dating record.

*Thank you.*

207:  The peak-trough smoothing observed in PDFs is predominantly controlled by some combination of age error and data density.  It's not clear to me that sampling from a wider area makes this any worse than from a single location, unless the relevant processes are operating at significantly different times in different regions under consideration.

*The UK covers a large enough area to represent a range of climates at any point in time, which is already well demonstrated by the BIIS reconstructions we use for our comparisons. Even at times of maximum extent, ice sheets are present in the north of the islands but not the south. At other times, ice sheets are only present in mountainous regions. As ice cover is the focus of this work we feel our sentence is suitable.*

209:  "Statistically significant" has formal meaning difficult to associate with a decay-corrected PDF curve.  I would suggest just "significant."

*Change made.*

Page 15, fig. 6: The authors appear to have incorrectly recorded uncorrected TIMS ages as being corrected TIMS ages for Dream Cave, shown here at 53.07N.  The authors of the cited source paper did also supply corrected ages, of about 2 - 3 kyr younger than shown here for these two samples.  Note they didn't propagate any uncertainty associated with the correction (not doing so was common practice at the time).

*This is correct, and we have used the corrected ages in the updated version of Figure 6. We also updated the dataset and adjusted Figure 5, 7 and 9. Many thanks for your vigilance!*

524.  Lechleitner et al 2021, appears to be an orphaned reference.

*Thank you for noticing, we have removed this reference.*